# `MobILE`: Model-Based Imitation Learning From Observation Alone

**Rahul Kidambi**[*]
Amazon Search & AI
Berkeley CA 94704.
`rk773@cornell.edu`

**Jonathan D. Chang**
CS Department, Cornell University
Ithaca NY 14853.
`jdc396@cornell.edu`

**Wen Sun**
CS Department, Cornell University
Ithaca NY 14853.
`ws455@cornell.edu`

## Abstract

This paper studies Imitation Learning from Observations alone (ILFO) where the learner is presented with expert demonstrations that consist only of states visited by an expert (without access to actions taken by the expert). We present a provably efficient model-based framework `MobILE` to solve the ILFO problem. `MobILE` involves carefully trading off strategic exploration against imitation - this is achieved by integrating the idea of optimism in the face of uncertainty into the distribution matching imitation learning (IL) framework. We provide a unified analysis for `MobILE`, and demonstrate that `MobILE` enjoys strong performance guarantees for classes of MDP dynamics that satisfy certain well studied notions of structural complexity. We also show that the ILFO problem is *strictly harder* than the standard IL problem by presenting an exponential sample complexity separation between IL and ILFO. We complement these theoretical results with experimental simulations on benchmark OpenAI Gym tasks that indicate the efficacy of `MobILE`. Code for implementing the `MobILE` framework is available at `https://github.com/rahulkidambi/MobILE-NeurIPS2021`.

## 1 Introduction

This paper considers *Imitation Learning from Observation Alone (ILFO)*. In ILFO, the learner is presented with sequences of states encountered by the expert, *without* access to the actions taken by the expert, meaning approaches based on a reduction to supervised learning (e.g., Behavior cloning (BC) [49], DAgger [50]) are not applicable. ILFO is more general and has potential for applications where the learner and expert have different action spaces, applications like sim-to-real [56, 14] etc.

Recently, [59] reduced the ILFO problem to a sequence of one-step distribution matching problems that results in obtaining a non-stationary policy. This approach, however, is sample inefficient for longer horizon tasks since the algorithm does not effectively reuse previously collected samples when solving the current sub-problem. Another line of work considers model-based methods to infer the expert's actions with either an inverse dynamics [63] or a forward dynamics [16] model; these recovered actions are then fed into an IL approach like BC to output the final policy. These works rely on stronger assumptions that are only satisfied for Markov Decision Processes (MDPs) with injective transition dynamics [68]; we return to this in the related works section.

---

[*]Work initiated when RK was a post-doc at Cornell University; work done outside Amazon.

35th Conference on Neural Information Processing Systems (NeurIPS 2021), virtual.

We introduce MobILE—Model-based Imitation Learning and Exploring, a model-based framework, to solve the ILFO problem. In contrast to existing model-based efforts, MobILE learns the forward transition dynamics model—a quantity that is well defined for any MDP. Importantly, MobILE *combines strategic exploration with imitation* by interleaving a model learning step with a bonus-based, optimistic distribution matching step – a perspective, to the best of our knowledge, that has not been considered in Imitation Learning. MobILE has the ability to automatically trade-off exploration and imitation. It simultaneously explores to collect data to refine the model and imitates the expert wherever the learned model is accurate and certain. At a high level, our theoretical results and experimental studies demonstrate that *systematic exploration is beneficial for solving ILFO reliably and efficiently,* and *optimism* is a both theoretically sound and practically effective approach

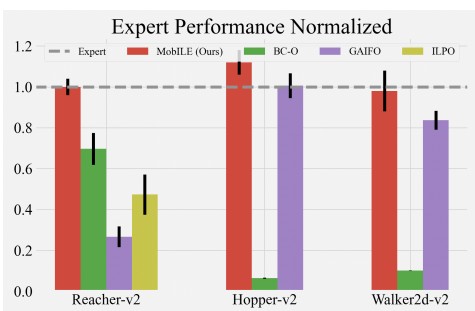

Figure 1: Expert performance normalized scores of ILFO algorithms averaged across 5 seeds in environments with discrete action spaces (Reacher-v2) and continuous action spaces (Hopper-v2 and Walker2d-v2).

for strategic exploration in ILFO (see Figure 1 for comparisons with other ILFO algorithms). This paper extends the realm of partial information problems (e.g. Reinforcement Learning and Bandits) where optimism has been shown to be crucial in obtaining strong performance, both in theory (e.g., $E^3$ [30], UCB [3]) and practice (e.g., RND [10]). This paper proves that incorporating optimism into the min-max IL framework [69, 22, 59] is *beneficial* for both the theoretical foundations and empirical performance of ILFO.

**Our Contributions:** We present MobILE (Algorithm 1), a provably efficient, model-based framework for ILFO that offers competitive results in benchmark gym tasks. MobILE can be instantiated with various implementation choices owing to its modular design. This paper's contributions are:

1. The MobILE framework combines ideas of model-based learning, optimism for exploration, and adversarial imitation learning. MobILE achieves global optimality with near-optimal regret bounds for classes of MDP dynamics that satisfy certain well studied notions of complexity. The key idea of MobILE is to use optimism to *trade-off imitation and exploration*.

2. We show an exponential sample complexity gap between ILFO and classic IL where one has access to expert's actions. This indicates that ILFO is *fundamentally harder* than IL. Our lower bound on ILFO also indicates that to achieve near optimal regret, one needs to perform systematic exploration rather than random or no exploration, both of which will incur sub-optimal regret.

3. We instantiate MobILE with a model ensemble of neural networks and a disagreement-based bonus. We present experimental results on benchmark OpenAI Gym tasks, indicating MobILE compares favorably to or outperforms existing approaches. Ablation studies indicate that optimism indeed helps in significantly improving the performance in practice.

### 1.1 Related Works

**Imitation Learning** (IL) is considered through the lens of two types of approaches: (a) behavior cloning (BC) [45] which casts IL as a reduction to supervised or full-information online learning [49, 50], or, (b) (adversarial) inverse RL [40, 1, 69, 17, 22, 29, 18], which involves minimizing various distribution divergences to solve the IL problem, either with the transition dynamics known (e.g., [69]), or unknown (e.g., [22]). MobILE does not assume knowledge of the transition dynamics, is model-based, and operates without access to the expert's actions.

**Imitation Learning from Observation Alone** (ILFO) [59] presents a model-free approach FAIL that outputs a non-stationary policy by reducing the ILFO problem into a sequence of min-max problems, one per time-step. While being theoretically sound, this approach cannot share data across different time steps and thus is not data efficient for long horizon problems. Also FAIL in theory only works for discrete actions. In contrast, our paper learns a stationary policy using model-based approaches by reusing data across all time steps and extends to continuous action space. Another line of work [63, 16, 66] relies on learning an estimate of expert action, often through the use of an inverse dynamics models, $P^e(a|s, s')$. Unfortunately, an inverse dynamics model is not well defined

in many benign problem instances. For instance, [68, remark 1, section 9.3] presents an example showing that inverse dynamics isn't well defined except in the case when the MDP dynamics is injective (i.e., no two actions could lead to the same next state from the current state. Note that even deterministic transition dynamics doesn't imply injectivity of the MDP dynamics). Furthermore, ILPO [16] applies to MDPs with deterministic transition dynamics and discrete actions. MobILE, on the other hand, learns the forward dynamics model which is always unique and well-defined for both deterministic and stochastic transitions and works with discrete and continuous actions. Another line of work in ILFO revolves around using hand-crafted cost functions that may rely on task-specific knowledge [44, 4, 53]. The performance of policy outputted by these efforts relies on the quality of the engineered cost functions. In contrast, MobILE does not require cost function engineering.

**Model-Based RL** has seen several advances [61, 36, 13] including ones based on deep learning (e.g., [34, 19, 38, 24, 37, 65]). Given MobILE's modularity, these advances in model-based RL can be translated to improved algorithms for the ILFO problem. MobILE bears parallels to provably efficient model-based RL approaches including $E^3$ [31, 27], R-MAX [7], UCRL [23], UCBVI [5], Linear MDP [67], LC$^3$ [25], Witness rank [58] which utilize optimism based approaches to trade-off exploration and exploitation. Our work utilizes optimism to trade-off *exploration and imitation*.

## 2 Setting

We consider episodic finite-horizon MDP $\mathcal{M} = \{\mathcal{S}, \mathcal{A}, P^\star, H, c, s_0\}$, where $\mathcal{S}, \mathcal{A}$ are the state and action space, $P^\star : \mathcal{S} \times \mathcal{A} \mapsto \mathcal{S}$ is the MDP's transition kernel, H is the horizon, $s_0$ is a fixed initial state (note that our work generalizes when we have a distribution over initial states), and $c$ is the *state-dependent* cost function $c : \mathcal{S} \mapsto [0, 1]$. Our result can be extended to the setting where $c : \mathcal{S} \times \mathcal{S} \mapsto [0, 1]$, i.e., the ground truth cost $c(s, s')$ depends on state and next state pairs. For analysis simplicity, we focus on $c : \mathcal{S} \mapsto [0, 1]$.[2]

We denote $d_P^\pi \in \Delta(\mathcal{S} \times \mathcal{A})$ as the average state-action distribution of policy $\pi$ under the transition kernel $P$, i.e., $d_P^\pi(s, a) := \frac{1}{H} \sum_{t=1}^{H} Pr(s_t = s, a_t = a | s_0, \pi, P)$, where $Pr(s_t = s, a_t = a | s_0, \pi, P)$ is the probability of reaching $(s, a)$ at time step $t$ starting from $s_0$ by following $\pi$ under transition kernel $P$. We abuse notation and write $s \sim d_P^\pi$ to denote a state $s$ is sampled from the state-wise distribution which marginalizes action over $d_P^\pi(s, a)$, i.e., $d_P^\pi(s) := \frac{1}{H} \sum_{t=1}^{H} Pr(s_t = s | s_0, \pi, P)$. For a given cost function $f : \mathcal{S} \mapsto [0, 1]$, $V_{P,f}^\pi$ denotes the expected total cost of $\pi$ under transition $P$ and cost function $f$. Similar to IL setting, in ILFO, the *ground truth cost $c$ is unknown*. Instead, we can query the expert, denoted as $\pi^e : \mathcal{S} \mapsto \Delta(\mathcal{A})$. Note that the expert $\pi^e$ could be stochastic and does not have to be the optimal policy. The expert, when queried, provides state-only demonstrations $\tau = \{s_0, s_1 \ldots s_H\}$, where $s_{t+1} \sim P^\star(\cdot | s_t, a_t)$ and $a_t \sim \pi^e(\cdot | s_t)$.

The goal is to leverage expert's state-wise demonstrations to learn a policy $\pi$ that performs as well as $\pi^e$ in terms of optimizing the ground truth cost $c$, with polynomial sample complexity on problem parameters such as horizon, number of expert samples and online samples and underlying MDP's complexity measures (see section 4 for precise examples). We track the progress of any (randomized) algorithm by measuring the (expected) regret incurred by a policy $\pi$ defined as $E[V^\pi] - V^{\pi^*}$ as a function of number of online interactions utilized by the algorithm to compute $\pi$.

### 2.1 Function Approximation Setup

Since the ground truth cost $c$ is unknown, we utilize the notion of a function class (i.e., discriminators) $\mathcal{F} \subset \mathcal{S} \mapsto [0, 1]$ to define the costs that can then be utilized by a planning algorithm (e.g. NPG [26]) for purposes of distribution matching with expert states. If the ground truth $c$ depends $(s, s')$, we use discriminators $\mathcal{F} \subset \mathcal{S} \times \mathcal{S} \mapsto [0, 1]$. Furthermore, we use a model class $\mathcal{P} \subset \mathcal{S} \times \mathcal{A} \mapsto \Delta(\mathcal{S})$ to capture the ground truth transition $P^\star$. For the theoretical results in the paper, we assume realizability:

**Assumption 1.** *Assume $\mathcal{F}$ and $\mathcal{P}$ captures ground truth cost and transition, i.e., $c \in \mathcal{F}$, $P^\star \in \mathcal{P}$.*

We will use Integral probability metric (IPM) with $\mathcal{F}$ as our divergence measure. Note that if $c \in \mathcal{F}$ and $c : \mathcal{S} \mapsto [0, 1]$, then IPM defined as $\max_{f \in \mathcal{F}} \mathbb{E}_{s \sim d^\pi} f(s) - \mathbb{E}_{s \sim d^{\pi^e}} f(s)$ directly upper

---

[2]Without any additional assumptions, in ILFO, learning to optimize action-dependent cost $c(s, a)$ (or $c(s, a, s')$ is **not possible**. For example, if there are two sequences of actions that generate the same sequence of states, without seeing expert's preference over actions, we do not know which actions to commit to.

---

**Algorithm 1** MobILE: The framework of **Mo**del-**b**ased **I**mitation **L**earning and **E**xploring for ILFO

---

1: **Require**: IPM class $\mathcal{F}$, dynamics model class $\mathcal{P}$, policy class $\Pi$, bonus function class $\mathcal{B}$, expert dataset $\mathcal{D}_e \equiv \{s_i^e\}_{i=1}^N$.
2: Initialize policy $\pi_0 \in \Pi$, replay buffer $\mathcal{D}_{-1} = \emptyset$.
3: **for** $t = 0, \cdots, T - 1$ **do**
4:     Execute $\pi_t$ in true environment $P^\star$ to get samples $\tau_t = \{s_k, a_k\}_{k=0}^{H-1} \cup s_H$. Append to replay buffer $\mathcal{D}_t = \mathcal{D}_{t-1} \cup \tau_t$.
5:     Update model and bonus: $\widehat{P}_{t+1} : \mathcal{S} \times \mathcal{A} \to \mathcal{S}$ and $b_{t+1} : \mathcal{S} \times \mathcal{A} \to \mathbb{R}^+$ using buffer $\mathcal{D}_t$.
6:     Optimistic model-based min-max IL: obtain $\pi_{t+1}$ by solving equation (1) with $\widehat{P}_{t+1}, b_{t+1}, \mathcal{D}_e$.
7: **end for**
8: **Return** $\pi_T$.

---

bounds sub-optimality gap $V^\pi - V^{\pi^e}$, where $V^\pi$ is the expected total cost of $\pi$ under cost function $c$. This justifies why minimizing IPM between two state distributions suffices [22, 59]. Similarly, if $c$ depends on $s, s'$, we can simply minimize IPM between two state-next state distributions, i.e., $\max_f \mathbb{E}_{s,s' \sim d^\pi} f(s, s') - \mathbb{E}_{s,s' \sim d^{\pi^e}} f(s, s')$ where discriminators now take $(s, s')$ as input.[3]

To permit generalization, we require $\mathcal{P}$ to have bounded complexity. For analytical simplicity, we assume $\mathcal{F}$ is discrete (but exponentially large), and we require the sample complexity of any PAC algorithm to scale polynomially with respect to its complexity $\ln(|\mathcal{F}|)$. The $\ln |\mathcal{F}|$ complexity can be replaced to bounded conventional complexity measures such as Rademacher complexity and covering number for continuous $\mathcal{F}$ (e.g., $\mathcal{F}$ being a Reproducing Kernel Hilbert Space).

## 3 Algorithm

We introduce MobILE (Algorithm 1) for the ILFO problem. MobILE utilizes (a) a function class $\mathcal{F}$ for Integral Probability Metric (IPM) based distribution matching, (b) a transition dynamics model class $\mathcal{P}$ for model learning, (c) a bonus parameterization $\mathcal{B}$ for exploration, (d) a policy class $\Pi$ for policy optimization. At every iteration, MobILE (in Algorithm 1) performs the following steps:

1. **Dynamics Model Learning:** execute policy in the environment online to obtain state-action-next state $(s, a, s')$ triples which are appended to the buffer $\mathcal{D}$. Fit a transition model $\widehat{P}$ on $\mathcal{D}$.

2. **Bonus Design:** design bonus to incentivize exploration where the learnt dynamics model is uncertain, i.e. the bonus $b(s, a)$ is large at state $s$ where $\widehat{P}(\cdot|s, a)$ is uncertain in terms of estimating $P^\star(\cdot|s, a)$, while $b(s, a)$ is small where $\widehat{P}(\cdot|s, a)$ is certain.

3. **Imitation-Exploration tradeoff:** Given discriminators $\mathcal{F}$, model $\widehat{P}$, bonus $b$ and expert dataset $\mathcal{D}_e$, perform distribution matching by solving the model-based IPM objective with bonus:

$$\pi_{t+1} \leftarrow \arg\min_{\pi \in \Pi} \max_{f \in \mathcal{F}} L(\pi, f; \widehat{P}, b, \mathcal{D}_e) := \mathbb{E}_{(s,a) \sim d_{\widehat{P}}^\pi} [f(s) - b(s, a)] - \mathbb{E}_{s \sim \mathcal{D}_e} [f(s)], \quad (1)$$

where $\mathbb{E}_{s \sim \mathcal{D}_e} f(s) := \sum_{s \in \mathcal{D}_e} f(s)/|\mathcal{D}_e|$.

Intuitively, the bonus cancels out discriminator's power in parts of the state space where the dynamics model $\widehat{P}$ is not accurate, thus offering freedom for MobILE to explore. We first explain MobILE's components and then discuss MobILE's key property—which is to trade-off *exploration and imitation*.

### 3.1 Components of MobILE

This section details MobILE's components.

**Dynamics model learning:** For the model fitting step in line 5, we assume that we get a calibrated model in the sense that: $\|\widehat{P}_t(\cdot|s, a) - P^\star(\cdot|s, a)\|_1 \leq \sigma_t(s, a), \forall s, a$ for some uncertainty measure $\sigma_t(s, a)$, similar to model-based RL works, e.g. [12]. We discuss ways to estimate $\sigma_t(s, a)$ in the bonus estimation below. There are many examples (discussed in Section 4) that permit efficient

---

[3]we slightly abuse notation here and denote $d^\pi$ as the average state-next state distribution of $\pi$, i.e., $d^\pi(s, s') := d^\pi(s) \int_a \pi(a|s) da P^\star(s'|s, a)$.

estimation of these quantities including tabular MDPs, Kernelized nonlinear regulator, nonparametric model such as Gaussian Processes. Consider a general function class $\mathcal{G} \subset \mathcal{S} \times \mathcal{A} \mapsto \mathcal{S}$, one can learn $\widehat{g}_t$ via solving a regression problem, i.e.,

$$\widehat{g}_t = \operatorname*{argmin}_{g \in \mathcal{G}} \sum_{s,a,s' \in \mathcal{D}_t} \|g(s,a) - s'\|_2^2, \tag{2}$$

and setting $\widehat{P}_t(\cdot|s,a) = \mathcal{N}\left(\widehat{g}_t(s,a), \sigma^2 I\right)$, where, $\sigma$ is the standard deviation of error induced by $\widehat{g}_t$. In practice, such parameterizations have been employed in several settings in RL with $\mathcal{G}$ being a multi-layer perceptron (MLP) based function class (e.g.,[48]). In Section 4, we also connect this with prior works in provable model-based RL literature.

**Bonus:** We utilize bonuses as a means to incentivize the policy to efficiently explore unknown parts of the state space for improved model learning (and hence better distribution matching). With the uncertainty measure $\sigma_t(s,a)$ obtained from calibrated model fitting, we can simply set the bonus $b_t(s,a) = O(H\sigma_t(s,a))$. How do we obtain $\sigma_t(s,a)$ in practice? For a general class $\mathcal{G}$, given the least square solution $\widehat{g}_t$, we can define a version space $\mathcal{G}_t$ as: $\mathcal{G}_t = \left\{g \in \mathcal{G} : \sum_{i=0}^{t-1} \sum_{h=0}^{H-1} \|g(s_h^t, a_h^t) - \widehat{g}_t(s_h^t, a_h^t)\|_2^2 \leq z_t\right\}$, with $z_t$ being a hyper parameter. The version space $\mathcal{G}_t$ is an *ensemble of functions* $g \in \mathcal{G}$ which has training error on $\mathcal{D}_t$ almost as small as the training error of the least square solution $\widehat{g}_t$. In other words, version space $\mathcal{G}_t$ contains functions that agree on the training set $\mathcal{D}_t$. The uncertainty measure at $(s,a)$ is then the *maximum disagreement* among models in $\mathcal{G}_t$, with $\sigma_t(s,a) \propto \sup_{f_1,f_2 \in \mathcal{G}_t} \|f_1(s,a) - f_2(s,a)\|_2$. Since $g \in \mathcal{G}_t$ agree on $\mathcal{D}_t$, a large $\sigma_t(s,a)$ indicates $(s,a)$ is novel. See example 3 for more theoretical details.

Empirically, disagreement among an ensemble [41, 6, 11, 43, 37] is used for designing bonuses that incentivize exploration. We utilize an neural network ensemble, where each model is trained on $\mathcal{D}_t$ (via SGD on squared loss Eq. 2) with different initialization. This approximates the version space $\mathcal{G}_t$, and the bonus is set as a function of maximum disagreement among the ensemble's predictions.

**Optimistic model-based min-max IL:** For model-based imitation (line 6), MobILE takes the current model $\widehat{P}_t$ and the discriminators $\mathcal{F}$ as inputs and performs policy search to minimize the divergence defined by $\widehat{P}_n$ and $\mathcal{F}$: $d_t(\pi, \pi^e) := \max_{f \in \mathcal{F}} \left[\mathbb{E}_{s,a \sim d_{\widehat{P}_t}^\pi} (f(s) - b_t(s,a)) - \mathbb{E}_{s \sim d^{\pi^e}} f(s)\right]$. Note that, for a fixed $\pi$, the $\arg\max_{f \in \mathcal{F}}$ is identical with or without the bonus term, since $\mathbb{E}_{s,a \sim d_{\widehat{P}_t}^\pi} b_t(s,a)$ is independent of $f$. In our implementation, we use the Maximum Mean Discrepancy (MMD) with a Radial Basis Function (RBF) kernel to model discriminators $\mathcal{F}$.[4] We compute $\operatorname{argmin}_\pi d_t(\pi, \pi^e)$ by iteratively (1) computing the $\operatorname{argmax}$ discriminator $f$ given the current $\pi$, and (2) using policy gradient methods (e.g., TRPO) to update $\pi$ inside $\widehat{P}_t$ with $f - b_t$ as the cost. Specifically, to find $\pi_t$ (line 6), we iterate between the following two steps:

1. Cost update: $\hat{f} = \operatorname*{argmax}_{f \in \mathcal{F}} \mathbb{E}_{s \sim d_{\widehat{P}_t}^{\hat\pi}} f(s) - \mathbb{E}_{s \sim \mathcal{D}^e} f(s)$,    2. PG Step: $\hat\pi = \hat\pi - \eta \cdot \nabla_\pi V_{\widehat{P}_t, \hat{f} - b_t}^{\hat\pi}$,

where the PG step uses the learnt dynamics model $\widehat{P}_t$ and the optimistic IPM cost $\hat{f}(s) - b_t(s,a)$. Note that for MMD, the cost update step has a closed-form solution.

## 3.2 Exploration And Imitation Tradeoff

We note that MobILE is performing an automatic *trade-off between exploration and imitation*. More specifically, the bonus is designed such that it has high values in the state space that have not been visited, and low values in the state space that have been frequently visited by the sequence of learned policies so far. Thus, by incorporating the bonus into the discriminator $f \in \mathcal{F}$ (e.g., $\widetilde{f}(s,a) = f(s) - b_t(s,a)$), we diminish the power of discriminator $f$ at novel state-action space regions, which relaxes the state-matching constraint (as the bonus cancels the penalty from the discriminators) at those novel regions so that exploration is encouraged. For well explored states, we force the learner's states to match the expert's using the full power of the discriminators. Our work uses optimism (via coupling bonus and discriminators) to carefully balance imitation and exploration.

---

[4]For MMD with kernel $k$, $\mathcal{F} = \{w^\top \phi(s,a) | \|w\|_2 \leq 1\}$ where $\phi$: $\langle \phi(s,a), \phi(s',a') \rangle = k((s,a), (s',a'))$.

# 4 Analysis

This section presents a general theorem for `MobILE` that uses the notion of *information gain* [57], and then specializes this result to common classes of stochastic MDPs such as discrete (tabular) MDPs, Kernelized nonlinear regulator [28], and general function class with bounded Eluder dimension [51].

Recall, Algorithm 1 generates one state-action trajectory $\tau^t := \{s_h^t, a_h^t\}_{h=0}^H$ at iteration $t$ and estimates model $\widehat{P}_t$ based on $\mathcal{D}_t = \tau^0, \dots, \tau^{t-1}$. We present our theorem under the assumption that model fitting gives us a model $\widehat{P}$ and a confidence interval of the model's prediction.

**Assumption 2** (Calibrated Model). *For all iteration $t$ with $t \in \mathbb{N}$, with probability $1 - \delta$, we have a model $\widehat{P}_t$ and its associated uncertainty measure $\sigma_t : \mathcal{S} \times \mathcal{A} \mapsto \mathbb{R}^+$, such that for all $s, a \in \mathcal{S} \times \mathcal{A}$[5]*

$$\left\| \widehat{P}_t(\cdot|s,a) - P^\star(\cdot|s,a) \right\|_1 \leq \min\left\{\sigma_t(s,a), 2\right\}.$$

Assumption 2 has featured in prior works (e.g., [12]) to prove regret bounds in model-based RL. Below we demonstrate examples that satisfy the above assumption.

**Example 1** (Discrete MDPs). *Given $\mathcal{D}_t$, denote $N(s,a)$ as the number of times $(s,a)$ appears in $\mathcal{D}_t$, and $N(s,a,s')$ number of times $(s,a,s')$ appears in $\mathcal{D}_t$. We can set $\widehat{P}_t(s'|s,a) = N(s,a,s')/N(s,a), \forall s, a, s'$. We can set $\sigma_t(s,a) = \widetilde{O}\left(\sqrt{S/N(s,a)}\right)$.*

**Example 2** (KNRs [28]). *For KNR, we have $P^\star(\cdot|s,a) = \mathcal{N}\left(W^\star\phi(s,a), \sigma^2 I\right)$ where feature mapping $\phi(s,a) \in \mathbb{R}^d$ and $\|\phi(s,a)\|_2 \leq 1$ for all $s, a$.[6] We can learn $\widehat{P}_t$ via Kernel Ridge regression, i.e., $\widehat{g}_t(s,a) = \widehat{W}_t\phi(s,a)$ where*

$$\widehat{W}_t = \operatorname*{argmin}_W \sum_{s,a,s' \in \mathcal{D}_t} \|W\phi(s,a) - s'\|_2^2 + \lambda \|W\|_F^2$$

*where $\|\cdot\|_F$ is the Frobenius norm. The uncertainty measure $\sigma_t(s,a) = \frac{\beta_t}{\sigma}\|\phi(s,a)\|_{\Sigma_t^{-1}}$, $\beta_t = \{2\lambda\|W^\star\|_2^2 + 8\sigma^2 \cdot [d_s \ln(5) + 2\ln(t^2/\delta) + \ln(4) + \ln(\det(\Sigma_t)/\det(\lambda I))]\}^{1/2}$, and, $\Sigma_t = \sum_{k=0}^{t-1} \sum_{h=1}^{H-1} \phi(s_h^k, a_h^k)\phi(s_h^k, a_h^k)^\top + \lambda I$ with $\lambda > 0$. See Proposition 12 for more details.*

Similar to RKHS, Gaussian processes (GPs) offers a calibrated model [57]. Note that GPs offer similar regret bounds as RKHS; so we do not discuss GPs and instead refer readers to [12].

**Example 3** (General class $\mathcal{G}$). *In this case, assume we have $P^\star(\cdot|s,a) = \mathcal{N}(g^\star(s,a), \sigma^2 I)$ with $g^\star \in \mathcal{G}$. Assume $\mathcal{G}$ is discrete (but could be exponentially large with complexity measure, $\ln(|\mathcal{G}|)$), and $\sup_{g \in \mathcal{G}, s, a} \|g(s,a)\|_2 \leq G \in \mathbb{R}^+$. Suppose model learning step is done by least square: $\widehat{g}_t = \operatorname*{argmin}_{g \in \mathcal{G}} \sum_{k=0}^{t-1} \sum_{h=0}^{H-1} \|g(s_h^k, a_h^k) - s_{h+1}^k\|_2^2$. Compute a version space $\mathcal{G}_t = \left\{ g \in \mathcal{G} : \sum_{k=0}^{t-1} \sum_{h=0}^{H-1} \|g(s_h^k, a_h^k) - \widehat{g}_t(s_h^k, a_h^k)\|_2^2 \leq z_t \right\}$, where $z_t = 2\sigma^2 G^2 \ln(2t^2|\mathcal{G}|/\delta)$ and use this for uncertainty computation. In particular, set uncertainty $\sigma_t(s,a) = \frac{1}{\sigma} \max_{g_1 \in \mathcal{G}, g_2 \in \mathcal{G}} \|g_1(s,a) - g_2(s,a)\|_2$, i.e., the maximum disagreement between any two functions in the version space $\mathcal{G}_t$. Refer to Proposition 14 for more details.*

The maximum disagreement above motivates our practical implementation where we use an ensemble of neural networks to approximate the version space and use the maximum disagreement among the models' predictions as the bonus. We refer readers to Section 6 for more details.

## 4.1 Regret Bound

We bound regret with the quantity named *Information Gain $\mathcal{I}$* (up to some constant scaling factor) [57]:

$$\mathcal{I}_T := \max_{\text{Alg}} \mathbb{E}_{\text{Alg}} \left[ \sum_{t=0}^{T-1} \sum_{h=0}^{H-1} \min\left\{\sigma_t^2(s_h^t, a_h^t), 1\right\} \right], \tag{3}$$

---

[5]the uncertainty measure $\sigma_t(s,a)$ will depend on the input failure probability $\delta$, which we drop here for notational simplicity. When we introduce specific examples, we will be explicit about the dependence on the failure probability $\delta$ which usually is in the order of $\ln(1/\delta)$.

[6]The covariance matrix can be generalized to any PSD matrix with bounded condition number.

where Alg is any adaptive algorithm (thus including Algorithm 1) that maps from history before iteration $t$ to some policy $\pi_t \in \Pi$. After the main theorem, we give concrete examples for $\mathcal{I}_T$ where we show that $\mathcal{I}_T$ has extremely mild growth rate with respect to $T$ (i.e., logarithimic). Denote $V^\pi$ as the expected total cost of $\pi$ under the true cost function $c$ and the real dynamics $P^\star$.

**Theorem 3** (Main result). *Assume model learning is calibrated (i.e., Assumption 2 holds for all $t$) and Assumption 1 holds. In Algorithm 1, set bonus $b_t(s,a) := H \min\{\sigma_t(s,a), 2\}$. There exists a set of parameters, such that after running Algorithm 1 for $T$ iterations, we have:*

$$\mathbb{E}\left[\min_{t \in [0,...,T-1]} V^{\pi_t} - V^{\pi^e}\right] \leq O\left(\frac{H^{2.5}\sqrt{\mathcal{I}_T}}{\sqrt{T}} + H\sqrt{\frac{\ln(TH|\mathcal{F}|)}{N}}\right).$$

Appendix A contains proof of Theorem 3. This theorem indicates that as long as $\mathcal{I}_T$ grows sublinearly $o(T)$, we find a policy that is at least as good as the expert policy when $T$ and $N$ approach infinity. For any discrete MDP, KNR [28], Gaussian Processes models [57], and general $\mathcal{G}$ with bounded Eluder dimension ([52, 42]), we can show that the growth rate of $\mathcal{I}_T$ with respect to $T$ is mild.

**Corollary 4** (Discrete MDP). *For discrete MDPs, $\mathcal{I}_T = \widetilde{O}(HS^2A)$ where $S = |\mathcal{S}|$, $A = |\mathcal{A}|$. Thus:*

$$\mathbb{E}\left[\min_{t \in [0,...,T-1]} V^{\pi_t} - V^{\pi^e}\right] = \widetilde{O}\left(\frac{H^3 S\sqrt{A}}{\sqrt{T}} + H\sqrt{\frac{\ln(|\mathcal{F}|)}{N}}\right).$$

Note that Corollary 4 (proof in Appendix A.1) hold for *any* MDPs (not just injective MDPs) and any stochastic expert policy. The dependence on $A, T$ is tight (see lower bound in 4.2). Now we specialize Theorem 3 to continuous MDPs below.

**Corollary 5** (KNRs (Example 2)). *For simplicity, consider the finite dimension setting $\phi : \mathcal{S} \times \mathcal{A} \mapsto \mathbb{R}^d$. We can show that $\mathcal{I}_T = \widetilde{O}\left(Hd + Hdd_s + Hd^2\right)$ (see Proposition 13 for details), where $d$ is the dimension of the feature $\phi(s,a)$ and $d_s$ is the dimension of the state space. Thus, we have* [7]

$$\mathbb{E}\left[\min_{t \in [0,...,T-1]} V^{\pi_t} - V^{\pi^e}\right] = \widetilde{O}\left(\frac{H^3\sqrt{dd_s + d^2}}{\sqrt{T}} + H\sqrt{\frac{\ln(|\mathcal{F}|)}{N}}\right).$$

**Corollary 6** (General $\mathcal{G}$ with bounded Eluder dimension (Example 3)). *For general $\mathcal{G}$, assume that $\mathcal{G}$ has Eluder-dimension $d_E(\epsilon)$ (Definition 3 in [42]). Denote $d_E = d_E(1/TH)$. The information gain is upper bounded as $\mathcal{I}_T = O\left(Hd_E + d_E \ln(T^3 H|\mathcal{G}|)\ln(TH)\right)$ (see Proposition 16). Thus,*

$$\mathbb{E}\left[\min_{t \in [0,...,T-1]} V^{\pi_t} - V^{\pi^e}\right] = \widetilde{O}\left(\frac{H^3\sqrt{d_E \ln(TH|\mathcal{G}|)}}{\sqrt{T}} + H\sqrt{\frac{\ln(|\mathcal{F}|)}{N}}\right).$$

Thus as long as $\mathcal{G}$ has bounded complexity in terms of the Eluder dimension [52, 42], `MobILE` with the maximum disagreement-based optimism leads to near-optimal guarantees.

## 4.2 Exploration in ILFO and the Exponential Gap between IL and ILFO

To show the benefit of strategic exploration over random exploration in ILFO, we present a *novel* reduction of the ILFO problem to a bandit optimization problem, for which strategic exploration is known to be *necessary* [9] for optimal bounds while random exploration is suboptimal; this reduction indicates that benefit of strategic exploration for solving ILFO efficiently. This reduction also demonstrate that there exists an exponential gap in terms of sample complexity between ILFO and classic IL that has access to expert actions. We leave the details of the reduction framework in Appendix A.4. The reduction allows us to derive the following lower bound for any ILFO algorithm.

**Theorem 7.** *There exists an MDP with number of actions $A \geq 2$, such that even with infinitely many expert data, any ILFO algorithm must occur expected commutative regret $\Omega(\sqrt{AT})$.*

Specifically we rely on the following reduction where solving ILFO, with even infinite expert data, is at least as hard as solving an MAB problem with the known optimal arm's mean reward which itself

---

[7] We use $\widetilde{O}$ to suppress log term except the $\ln(|\mathcal{G}|)$ and $\ln(|\mathcal{F}|)$ which present the complexity of $\mathcal{F}$ and $\mathcal{G}$.

occurs the same worst case $\sqrt{AT}$ cumulative regret bound as the one in the classic MAB setting. For MAB, it is known that random exploration such as $\epsilon$-greedy will occur suboptimal regret $O(T^{2/3})$. Thus to achieve optimal $\sqrt{T}$ rate, one needs to leverage strategic exploration (e.g., optimism).

Methods such as BC for IL have sample complexity that scales as poly $\ln(A)$, e.g., see [2, Theorem 14.3, Chapter 14] which shows that for tabular MDP, BC learns a policy whose performance is $O(H^2\sqrt{S\ln(A)/N})$ away from the expert's performance (here $S$ is the number of states in the tabular MDP). Similarly, in interactive IL setting, DAgger [50] can also achieve poly $\ln(A)$ dependence in sample complexity. The *exponential gap* in the sample complexity dependence on $A$ between IL and ILFO formalizes the additional difficulty encountered by learning algorithms in ILFO.

# 5 Practical Instantiation of `MobILE`

We present a brief practical instantiation `MobILE`'s components with details in Appendix Section C.
**Dynamics model learning:** We employ Gaussian Dynamics Models parameterized by an MLP [48, 32], i.e., $\widehat{P}(s,a) := \mathcal{N}(h_\theta(s,a), \sigma^2 I)$, where, $h_\theta(s,a) = s + \sigma_{\Delta_s} \cdot \text{MLP}_\theta(s_c, a_c)$, where, $\theta$ are MLP's trainable parameters, $s_c = (s - \mu_s)/\sigma_s$, $a_c = (a - \mu_a)/\sigma_a$ with $\mu_s, \mu_a$ (and $\sigma_s, \sigma_a$) being the mean of states, actions (and standard deviation of states and actions) in the replay buffer $\mathcal{D}$. Next, for $(s, a, s') \in \mathcal{D}$, $\Delta_s = s' - s$ and $\sigma_{\Delta_s}$ is the standard deviation of the state differences $\Delta_s \in \mathcal{D}$. We use SGD with momentum [60] for training the parameters $\theta$ of the MLP.
**Discriminator parameterization:** We utilize MMD as our choice of IPM and define the discriminator as $f(s) = w^\top \psi(s)$, where, $\psi(s)$ are Random Fourier Features [46].
**Bonus parameterization:** We utilize the discrepancy between predictions of a pair of dynamics models $h_{\theta_1}(s,a)$ and $h_{\theta_2}(s,a)$ for designing the bonus. Empirically, we found that using more than two models in the ensemble offered little to no improvements. Denote the disagreement at any $(s,a)$ as $\delta(s,a) = \|h_{\theta_1}(s,a) - h_{\theta_2}(s,a)\|_2$, and $\delta_\mathcal{D} = \max_{(s,a)\sim\mathcal{D}} \delta(s,a)$ is the max discrepancy of a replay buffer $\mathcal{D}$. We set bonus as $b(s,a) = \lambda \cdot \min(\delta(s,a)/\delta_\mathcal{D}$, where $\lambda > 0$ is a tunable parameter.
**PG oracle:** We use TRPO [54] to perform incremental policy optimization inside the learned model.

# 6 Experiments

This section seeks to answer the following questions: (1) How does `MobILE` compare against other benchmark algorithms? (2) How does optimism impact sample efficiency/final performance? (3) How does increasing the number of expert samples impact the quality of policy outputted by `MobILE`?

We consider tasks from Open AI Gym [8] simulated with Mujoco [62]: `Cartpole-v1`, `Reacher-v2`, `Swimmer-v2`, `Hopper-v2` and `Walker2d-v2`. We train an expert for each task using TRPO [54] until we obtain an expert policy of average value $460, -10, 38, 3000, 2000$ respectively. We setup `Swimmer-v2`, `Hopper-v2`,`Walker2d-v2` similar to prior model-based RL works [33, 39, 38, 48, 32].

We compare `MobILE` against the following algorithms: Behavior Cloning (BC), GAIL [22], BC-O [63], ILPO [16] (for environments with discrete actions), GAIFO [64]. Furthermore, recall that BC and GAIL utilize both expert states and actions, information that is not available for ILFO. This makes both BC and GAIL idealistic targets for comparing ILFO methods like `MobILE` against. As reported by Torabi et al. [63], BC outperforms BC-O in all benchmark results. Moreover, our results indicate `MobILE` outperforms GAIL and GAIFO in terms of sample efficiency. With reasonable amount of parameter tuning, BC serves as a very strong baseline and nearly solves *deterministic* Mujoco environments. We use code released by the authors for BC-O and ILPO. For GAIL we use an open source implementation [21], and for GAIFO, we modify the GAIL implementation as described by the authors. We present our results through (a) learning curves obtained by averaging the progress of the algorithm across 5 seeds, and, (b) bar plot showing expert normalized scores averaged across 5 seeds using the best performing policy obtained with each seed. Normalized score refers to ratio of policy's score over the expert score (so that expert has normalized score of 1). For `Reacher-v2`, since the expert policy has a negative score, we add an constant before normalization. More details can be found in Appendix C.

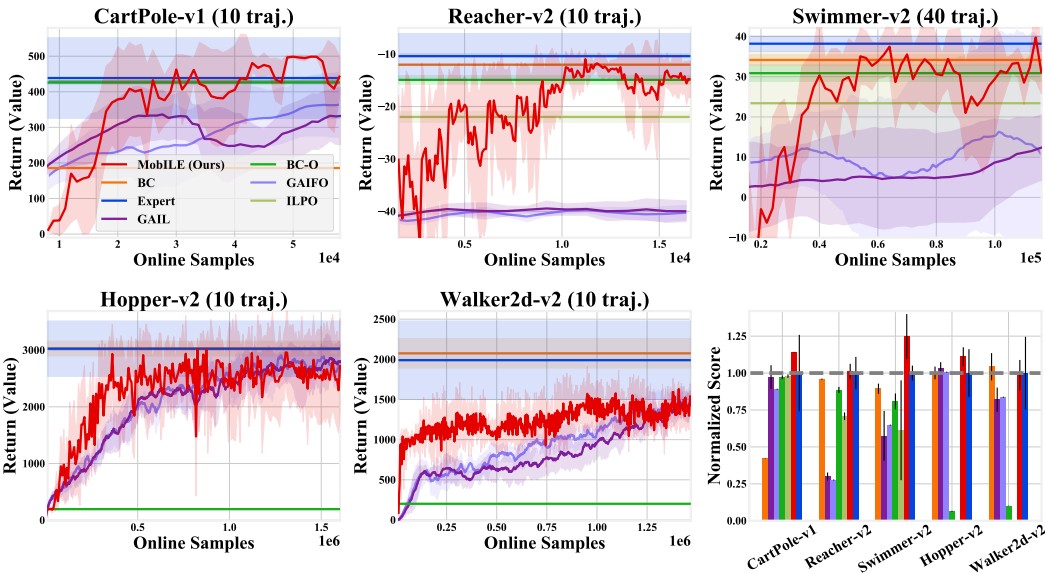

Figure 2: Comparing `MobILE` (red) against BC (orange), BC-O (green), GAIL (purple), GAIFO (periwinkle), ILPO (green olive). The learning curves are obtained by averaging all algorithms over 5 seeds. `MobILE` outperforms BC-O, GAIL and matches BC's behavior despite `MobILE` not having access to expert actions. The bar plot (bottom-right) presents the best performing policy outputted by each algorithm averaged across 5 seeds for each algorithm. `MobILE` clearly outperforms BC-O, GAIFO, ILPO while matching the behavior of IL algorithms like BC/GAIL which use expert actions.

## 6.1 Benchmarking `MobILE` on MuJoCo suite

Figure 2 compares `MobILE` with BC, BC-O, GAIL, GAIFO and ILPO. `MobILE` consistently matches or exceeds BC/GAIL's performance *despite BC/GAIL having access to actions taken by the expert* and `MobILE` functioning *without* expert action information. `MobILE`, also, consistently improves upon the behavior of ILFO methods such as BC-O, ILPO, and GAIFO. We see that BC does remarkably well in these benchmarks owing to determinism in the transition dynamics; in the appendix, we consider a variant of the cartpole environment with stochastic dynamics. Our results suggest that BC struggles with stochasticity in the dynamics and fails to solve this task, while `MobILE` continues to reliably solve this task. Also, note that we utilize 10 expert trajectories for all environments except `Swimmer-v2`; this is because all algorithms (including `MobILE`) present results with high variance. We include a learning curve for `Swimmer-v2` with 10 expert trajectories in the appendix. The bar plot in Figure 2 shows that within the sample budget shown in the learning curves, `MobILE` (being a model-based algorithm), presents superior performance in terms of matching expert, thus indicating it is more sample efficient than GAIFO, GAIL (both being model-free methods), ILPO and BC-O.

## 6.2 Importance of the optimistic MDP construction

Figure 3 presents results obtained by running `MobILE` with and without optimism. In the absence of optimism, the algorithm either tends to be sample inefficient in achieving expert performance or

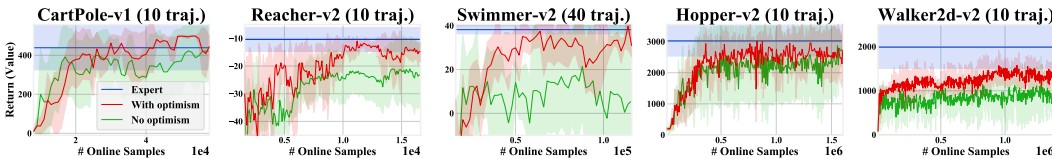

Figure 3: Learning curves obtained by running `MobILE` with (red) and without (green) optimism. Without optimism, the algorithm learns slowly or does not match the expert, whereas, with optimism, `MobILE` shows improved behavior by automatically trading off exploration and imitation.

completely fails to solve the problem. Note that without optimism, the algorithm isn't explicitly incentivized to explore – only implicitly exploring due to noise induced by sampling actions. This, however, is not sufficient to solve the problem efficiently. In contrast, `MobILE` with optimism presents improved behavior and in most cases, solves the environments with fewer online interactions.

### 6.3 Varying Number of Expert Samples

Table 1 shows the impact of increasing the number of samples drawn from the expert policy for solving the ILFO problem. The main takeaway is that increasing the number of expert samples aids `MobILE` in reliably solving the problem (i.e. with lesser variance).

Table 1: Expert normalized score and standard deviation of policy outputted by `MobILE` when varying number of expert trajectories as $E_1$ and $E_2$ (specific values represented in parentheses)

| Environment | $E_1$ | $E_2$ | Expert |
|---|---|---|---|
| Cartpole-v1 | $1.07 \pm 0.15$ (5) | $1.14 \pm 0$ (10) | $1 \pm 0.25$ |
| Reacher-v2 | $1.01 \pm 0.05$ (10) | $0.997 \pm 0.055$ (20) | $1 \pm 0.11$ |
| Swimmer-v2 | $1.54 \pm 1.1$ (10) | $1.25 \pm 0.15$ (40) | $1 \pm 0.05$ |
| Hopper-v2 | $1.11 \pm 0.064$ (10) | $1.16 \pm 0.03$ (40) | $1 \pm 0.16$ |
| Walker2d-v2 | $0.975 \pm 0.12$ (10) | $0.94 \pm 0.038$ (50) | $1 \pm 0.25$ |

## 7 Conclusions

This paper introduces `MobILE`, a model-based ILFO approach that is applicable to MDPs with stochastic dynamics and continuous action spaces. `MobILE` trades-off exploration and imitation, and this perspective is shown to be important for solving the ILFO efficiently both in theory and in practice. Future works include exploring other means for learning dynamics models, performing strategic exploration and extending `MobILE` to problems with rich observation spaces (e.g. videos).

By not even needing the actions to imitate, ILFO algorithms allow for learning algorithms to capitalize on large amounts of video data available online. Moreover, in ILFO, the learner is successful if it learns to imitate the expert. Any expert policy designed by bad actors can naturally lead to obtaining new policies that continue to imitate and be a negative influence to the society. With this perspective in mind, any expert policy must be thoroughly vetted in order to ensure ILFO algorithms including `MobILE` are employed in ways that benefit the society.

## Acknowledgements

Rahul Kidambi acknowledges funding from NSF TRIPODS Award CCF − 1740822 at Cornell University. All content represents the opinion of the authors, which is not necessarily shared or endorsed by their respective employers and/or sponsors.

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
