# Contents

# A   Analysis of Algorithm 1

We start by presenting the proof for the unified main result in Theorem 3. We then discuss the bounds for special instances individually.

The following lemma shows that under Assumption 2, with $b_t(s,a) = H \min\{\sigma_t(s,a), 2\}$, we achieve *optimism* at all iterations.

**Lemma 8** (Optimism). *Assume Assumption 2 holds, and set $b_t(s,a) = H \min\{\sigma_t(s,a), 2\}$. For all state-wise cost function $f : \mathcal{S} \mapsto [0,1]$, denote the bonus enhance cost as $\widetilde{f}_t(s,a) := f(s) - b_t(s,a)$. For all policy $\pi$, we have the following optimism:*

$$V^{\pi}_{\widehat{P}_t, \widetilde{f}_t} \leq V^{\pi}_{P,f}, \forall t.$$

*Proof.* In the proof, we drop subscript $t$ for notation simplicity. We consider a fixed function $f$ and policy $\pi$. Also let us denote $\widehat{V}^{\pi}$ as the value function of $\pi$ under $(\widehat{P}, \widetilde{f})$, and $V^{\pi}$ as the value function under $(P,f)$.

Let us start from $h = H$, where we have $\widehat{V}^{\pi}_H(s) = V^{\pi}_H(s) = 0$. Assume inductive hypothesis holds at $h+1$, i.e., for any $s,a$, we have $\widehat{Q}^{\pi}_{h+1}(s,a) \leq Q^{\pi}_{h+1}(s,a)$. Now let us move to $h$. We have:

$$
\begin{aligned}
\widehat{Q}^{\pi}_h(s,a) - Q^{\pi}_h(s,a) &= \widetilde{f}(s,a) + \mathbb{E}_{s' \sim \widehat{P}(\cdot|s,a)} \widehat{V}^{\pi}_{h+1}(s') - f(s) - \mathbb{E}_{s' \sim P(\cdot|s,a)} V^{\pi}_{h+1}(s') \\
&\leq -H \min\{\sigma(s,a), 2\} + \mathbb{E}_{s' \sim \widehat{P}(\cdot|s,a)} V^{\pi}_{h+1}(s') - \mathbb{E}_{s' \sim P(\cdot|s,a)} V^{\pi}_{h+1}(s') \\
&\leq -H \min\{\sigma(s,a), 2\} + H \left\| \widehat{P}(\cdot|s,a) - P(\cdot|s,a) \right\|_1 \\
&\leq -H \min\{\sigma(s,a), 2\} + H \min\{\sigma(s,a), 2\} = 0,
\end{aligned}
$$

where the first inequality uses the inductive hypothesis at time step $h+1$. Finally, note that $V^{\pi}_h(s) = \mathbb{E}_{a \sim \pi(s)} Q^{\pi}_h(s,a)$, which leads to $\widehat{V}^{\pi}_h(s) \leq V^{\pi}_h(s)$. This concludes the induction step.   □

The next lemma concerns the statistical error from finite sample estimation of $\mathbb{E}_{s \sim d^{\pi^e}} f(s)$.

**Lemma 9.** *Fix $\delta \in (0,1)$. For all $t$, we have that with probability at least $1 - \delta$,*

$$\left| \mathbb{E}_{s \sim d^{\pi^e}} f(s) - \sum_{i=1}^{N} f(s^e_i)/N \right| \leq 2\sqrt{\frac{\ln\left(2t^2|\mathcal{F}|/\delta\right)}{N}}, \forall f \in \mathcal{F}.$$

*Proof.* For any $t$, we set the failure probability to be $6\delta/(t^2\pi^2)$ at iteration $t$ where we abuse notation and point out that $\pi = 3.14159...$. Thus the total failure probability for all $t \in \mathbb{N}$ is at most $\delta$. We then apply classic Hoeffding inequality to bound $\mathbb{E}_{s \sim d^{\pi^e}} f(s) - \sum_{i=1}^{N} f(s^e_i)/N$ with the fact that $f(s) \in [0,1]$ for all $s$. We conclude the proof by taking a union bound over all $f \in \mathcal{F}$.   □

Note that here we have assumed $s^e_i \sim d^{\pi^e}$ is i.i.d sampled from $d^{\pi^e}$. This can easily be achieved by randomly sampling a state from each expert trajectory. Note that we can easily deal with i.i.d trajectories, i.e., if our expert data contains $N$ many i.i.d trajectories $\{\tau^1, \ldots, \tau^N\}$, we can apply concentration on the trajectory level, and get:

$$\left| \mathbb{E}_{\tau \sim \pi^e} \left[ \sum_{h=0}^{H-1} f(s_h) \right] - \frac{1}{N} \sum_{i=1}^{N} \sum_{h=0}^{H-1} f(s^i_h) \right| \leq O\left( H\sqrt{\frac{\ln(t^2|\mathcal{F}|/\delta)}{N}} \right),$$

where $\tau \sim \pi$ denotes that a trajectory $\tau$ being sampled based on $\pi$, $s_h^i$ denotes the state at time step $h$ on the i-th expert trajectory. Also note that we have $\mathbb{E}_{s\sim d^\pi} f(s) = \frac{1}{H}\mathbb{E}_{\tau\sim\pi}\left[\sum_{h=0}^{H-1} f(s_h)\right]$ for any $\pi, f$. Together this immediately implies that:

$$\left|\mathbb{E}_{s\sim d^{\pi^e}} f(s) - \frac{1}{NH}\sum_{i=1}^{N}\sum_{h=0}^{H-1} f(s_h^i)\right| \leq O\left(\sqrt{\frac{\ln(t^2|\mathcal{F}|/\delta)}{N}}\right),$$

which matches to the bound in Lemma 9.

Now we conclude the proof for Theorem 3.

*Proof of Theorem 3.* Assume that Assumption 2 and the event in Lemma 9 hold. Denote the joint of these two events as $\mathcal{E}$. Note that the probability of $\overline{\mathcal{E}}$ is at most $2\delta$. For notation simplicity, denote $\epsilon_{stats} = 2\sqrt{\frac{\ln(2T^2|\mathcal{F}|/\delta)}{N}}$.

In each model-based planning phase, recall that we perform model-based optimization on the following objective:

$$\pi_t = \operatorname*{argmin}_{\pi\in\Pi}\max_{f\in F}\left[\mathbb{E}_{s,a\sim d^\pi_{\widehat{P}_t}}[f(s) - b_t(s,a)] - \sum_{i=1}^{N} f(s_i^e)/N\right].$$

Note that for any $\pi$, using the inequality in Lemma 9, we have:

$$\max_{f\in\mathcal{F}_t}\left[\mathbb{E}_{s,a\sim d^\pi_{\widehat{P}_t}}(f(s) - b_t(s,a)) - \sum_{i=1}^{N} f(s_i^e)/N\right]$$

$$= \max_{f\in\mathcal{F}}\left[\mathbb{E}_{s,a\sim d^\pi_{\widehat{P}_t}}(f(s) - b_t(s,a)) - \mathbb{E}_{s\sim d^{\pi^e}} f(s) + \mathbb{E}_{s\sim d^{\pi^e}} f(s) - \sum_{i=1}^{N} f(s_i^e)/N\right]$$

$$\leq \max_{f\in\mathcal{F}}\left[\mathbb{E}_{s,a\sim d^\pi_{\widehat{P}_t}}(f(s) - b_t(s,a)) - \mathbb{E}_{s\sim d^{\pi^e}} f(s)\right] + \max_{f\in F}\left[\mathbb{E}_{s\sim d^{\pi^e}} f(s) - \sum_{i=1}^{N} f(s_i^e)/N\right]$$

$$\leq \max_{f\in\mathcal{F}}\left[\mathbb{E}_{s,a\sim d^\pi_{\widehat{P}_t}}(f(s) - b_t(s,a)) - \mathbb{E}_{s,a\sim d^{\pi^e}_{\widehat{P}_t}}(f(s) - b_t(s,a))\right] + \epsilon_{stats}$$

where in the last inequality we use optimism from Lemma 8, i.e., $\mathbb{E}_{s,a\sim d^{\pi^e}_{\widehat{P}_t}}(f(s) - b_t(s,a)) \leq \mathbb{E}_{s\sim d^{\pi^e}} f(s)$.

Hence, for $\pi_t$, since it is the minimizer and $\pi^e \in \Pi$, we must have:

$$\max_{f\in\mathcal{F}}\left[\mathbb{E}_{s,a\sim d^{\pi_t}_{\widehat{P}_t}}(f(s) - b_t(s,a)) - \sum_{i=1}^{N} f(s_i^e)/N\right]$$

$$\leq \max_{f\in\mathcal{F}}\left[\mathbb{E}_{s,a\sim d^{\pi^e}_{\widehat{P}_t}}(f(s) - b_t(s,a)) - \sum_{i=1}^{N} f(s_i^e)/N\right]$$

$$\leq \max_{f\in\mathcal{F}}\left[\mathbb{E}_{s,a\sim d^{\pi^e}_{\widehat{P}_t}}(f(s) - b_t(s,a)) - \mathbb{E}_{s,a\sim d^{\pi^e}_{\widehat{P}_t}}(f(s) - b_t(s,a))\right] + \epsilon_{stats} = \epsilon_{stats}.$$

Note that $\mathcal{F}$ contains $c$, we must have:

$$\mathbb{E}_{s,a\sim d^{\pi_t}_{\widehat{P}_t}}[c(s) - b_t(s,a)] \leq \sum_{i=1}^{N} c(s_i^e)/N + \epsilon_{stats} \leq \mathbb{E}_{s\sim d^{\pi^e}} c(s) + 2\epsilon_{stats},$$

which means that $V^{\pi_t}_{\widehat{P}_t;\widetilde{c}_t} \leq V^{\pi^e} + 2H\epsilon_{stats}$.

Now we compute the regret in episode $t$. First recall that $b_t(s,a) = H\min\{\sigma_t(s,a), 2\}$, which means that $\|b_t\|_\infty \leq 2H$ as $\|c\|_\infty \leq 1$, which means that $\|c - b_t\|_\infty \leq 2H$. Thus, $\left\|V^\pi_{\widehat{P};c-b_t}\right\|_\infty \leq 2H^2$. Recall simulation lemma (Lemma 18), we have:

$$V^{\pi_t} - V^{\pi^e} \leq V^{\pi_t} - V^{\pi_t}_{\widehat{P}_t;\widetilde{c}_t} + 2H\epsilon_{stats}$$

$$= H\mathbb{E}_{s,a\sim d^{\pi_t}}\left[|\widetilde{c}_t(s,a)-c(s)|+2H^2\left\|\widehat{P}_t(\cdot|s,a)-P^\star(\cdot|s,a)\right\|_1\right]+2H\epsilon_{stat}$$

$$= H\mathbb{E}_{s,a\sim d^{\pi_t}}\left[H\min\{\sigma_t(s,a),2\}+2H^2\left\|\widehat{P}_t(\cdot|s,a)-P^\star(\cdot|s,a)\right\|_1\right]+2H\epsilon_{stat}$$

$$\leq H\mathbb{E}_{s,a\sim d^{\pi_t}}\left[H\min\{\sigma_t(s,a),2\}+2H^2\min\{\sigma_t(s,a),2\}\right]+2H\epsilon_{stat}$$

$$\leq 3H^3\mathbb{E}_{s,a\sim d^{\pi_t}}\min\{\sigma_t(s,a),2\}+2H\epsilon_{stat}$$

$$\leq 6H^3\mathbb{E}_{s,a\sim d^{\pi_t}}\min\{\sigma_t(s,a),1\}+2H\epsilon_{stat}$$

Now sum over $t$, and denote $\mathbb{E}_{\pi_t}$ as the conditional expectation conditioned on the history from iteration 0 to $t-1$, we get:

$$\sum_{t=0}^{T-1}\left[V^{\pi_t}-V^{\pi^e}\right]\leq 6H^2\sum_{t=0}^{T-1}\mathbb{E}_{\pi_t}\left[\sum_{h=0}^{H-1}\min\{\sigma_t(s_h^t,a_h^t),1\}\right]+2HT\epsilon_{stat}$$

$$\leq 6H^2\sum_{t=0}^{T-1}\left[\sqrt{H}\sqrt{\mathbb{E}_{\pi_t}\sum_{h=0}^{H-1}\min\{\sigma_t^2(s_h^t,a_h^t),1\}}\right]+2HT\epsilon_{stat},$$

where in the last inequality we use $\mathbb{E}[a^\top b]\leq\sqrt{\mathbb{E}[\|a\|_2^2]\mathbb{E}[\|b\|_2^2]}$.

Recall that $\pi_t$ are random quantities, add expectation on both sides of the above inequality, and consider the case where $\mathcal{E}$ holds and $\overline{\mathcal{E}}$ holds, we have:

$$\mathbb{E}\left[\sum_{t=0}^{T-1}\left(V^{\pi_t}-V^{\pi^e}\right)\right]\leq 6H^{2.5}\mathbb{E}\left[\sum_{t=0}^{T-1}\sqrt{\mathbb{E}_{\pi_t}\sum_{h=0}^{H-1}\min\left\{\sigma_t^2(s_h^t,a_h^t),1\right\}}\right]+2HT\epsilon_{stat}+\mathbb{P}(\overline{\mathcal{E}})TH$$

$$\leq 6H^{2.5}\left[\sqrt{T}\sqrt{\mathbb{E}\left[\sum_{t=0}^{T-1}\sum_{h=0}^{H-1}\min\left\{\sigma_t^2(s_h^t,a_h^t),1\right\}\right]}\right]+2HT\epsilon_{stat}+2\delta TH,$$

where in the last inequality, we use $\mathbb{E}[a^\top b]\leq\sqrt{\mathbb{E}[\|a\|_2^2]\mathbb{E}[\|b\|_2^2]}$. This implies that that:

$$\mathbb{E}\left[\min_t V^{\pi_t}-V^{\pi^e}\right]\leq\frac{6H^{2.5}}{\sqrt{T}}\sqrt{\max_{\text{Alg}}\mathbb{E}_{\text{Alg}}\left[\sum_{t=0}^{T-1}\sum_{h=0}^{H-1}\min\left\{\sigma_t^2(s_h^t,a_h^t),1\right\}\right]}+2H\epsilon_{stats}+2H\delta.$$

Set $\delta=1/(HT)$, we get:

$$\mathbb{E}\left[V^\pi-V^{\pi^e}\right]\leq\frac{6H^{2.5}}{\sqrt{T}}\sqrt{\max_{\text{Alg}}\mathbb{E}_{\text{Alg}}\left[\sum_{t=0}^{T-1}\sum_{h=0}^{H-1}\min\left\{\sigma_t^2(s_h^t,a_h^t),1\right\}\right]}+2H\sqrt{\frac{\ln(T^3H|\mathcal{F}|)}{N}}+\frac{2}{T}$$

where Alg is any adaptive mapping that maps from history from $t=0$ to the end of the $t-1$ iteration to to some policy $\pi_t$. This concludes the proof. $\qquad\square$

Below we discuss special cases.

## A.1 Discrete MDPs

**Proposition 10** (Discrete MDP Bonus). *With $\delta\in(0,1)$. With probability at least $1-\delta$, for all $t\in\mathbb{N}$, we have:*

$$\left\|\widehat{P}_t(\cdot|s,a)-P^\star(\cdot|s,a)\right\|_1\leq\min\left\{\sqrt{\frac{S\ln(t^2SA/\delta)}{N_t(s,a)}},2\right\}.$$

*Proof.* The proof simply uses the concentration result for $\widehat{P}_t$ under the $\ell_1$ norm. For a fixed $t$ and $s,a$ pair, using Lemma 6.2 in [2], we have that with probability at least $1-\delta$,

$$\left\|\widehat{P}_t(\cdot|s,a)-P^\star(\cdot|s,a)\right\|_1\leq\sqrt{\frac{S\ln(1/\delta)}{N_t(s,a)}}.$$

Applying union bound over all iterations and all $s, a$ pairs, we conclude the proof. $\qquad\square$

What left is to bound the information gain $\mathcal{I}$ for the tabular case. For this, we can simply use the Proposition 13 that we develop in the next section for KNR. This is because in KNR, when we set the feature mapping $\phi(s, a) \in \mathbb{R}^{|\mathcal{S}||\mathcal{A}|}$ to be a one-hot vector with zero everywhere except one in the entry corresponding to $(s, a)$ pair, the information gain in KNR is reduced to the information gain in the tabular model.

**Proposition 11** (Information Gain in discrete MDPs). *We have:*

$$\mathcal{I}_T = O\left(HS^2 A \cdot \ln(TSA/\delta)\ln(1 + TH)\right).$$

*Proof.* Using Lemma B.6 in [25], we have:

$$\sum_{t=0}^{T-1} \min\left\{\sum_{h=0}^{H-1} \frac{1}{N_t(s_h^t, a_h^t)}, 1\right\} \leq 2SA \ln(1 + TH).$$

Now using the definition of information gain, we have:

$$\mathcal{I}_T = \sum_{t=0}^{T-1}\sum_{h=0}^{H-1} \min\left\{\sigma_t^2(s_h^t, a_h^t), 1\right\} \leq S \ln(T^2 SA/\delta)H \sum_{t=0}^{T-1} \min\left\{\sum_{h=0}^{H-1} \frac{1}{N_t(s_h^t, a_h^t)}, 1\right\}$$
$$\leq 2HS^2 A \ln(T^2 SA/\delta)\ln(1 + TH)$$

This concludes the proof. $\qquad\square$

## A.2 KNRs

Recall the KNR setting from Example 2. The following proposition shows that the bonus designed in Example 2 is valid.

**Proposition 12** (KNR Bonus). *Fix $\delta \in (0, 1)$. With probability at least $1 - \delta$, for all $t \in \mathbb{N}$, we have:*

$$\left\|\widehat{P}_t(\cdot|s, a) - P^\star(\cdot|s, a)\right\|_1 \leq \min\left\{\frac{\beta_t}{\sigma}\|\phi(s, a)\|_{\Sigma_t^{-1}}, 2\right\}, \forall s, a,$$

*where $\beta_t = \sqrt{2\lambda\|W^\star\|_2^2 + 8\sigma^2\left(d_s \ln(5) + 2\ln(t^2/\delta) + \ln(4) + \ln\left(\det(\Sigma_t)/\det(\lambda I)\right)\right)}$.*

*Proof.* The proof directly follows the confidence ball construction and proof from [25]. Specifically, from Lemma B.5 in [25], we have that with probability at least $1 - \delta$, for all $t$:

$$\left\|\left(\widehat{W}_t - W^\star\right)(\Sigma_t)^{1/2}\right\|_2^2 \leq \beta_t^2.$$

Thus, with Lemma 19, we have:

$$\left\|\widehat{P}_t(\cdot|s, a) - P^\star(\cdot|s, a)\right\|_1 \leq \frac{1}{\sigma}\left\|(\widehat{W}_t - W^\star)\phi(s, a)\right\|_2 \leq \left\|(\widehat{W}_t - W^\star)(\Sigma_t)^{1/2}\right\|\|\phi(s, a)\|_{\Sigma_t^{-1}}/\sigma \leq \frac{\beta_t}{\sigma}\|\phi(s, a)\|_{\Sigma_t^{-1}}.$$

This concludes the proof. $\qquad\square$

The following proposition bounds the information gain quantity.

**Proposition 13** (Information Gain on KNRs). *For simplicity, let us assume $\phi : \mathcal{S} \times \mathcal{A} \mapsto \mathbb{R}^d$, i.e., $\phi(s, a)$ is a $d$-dim feature vector. In this case, we will have:*

$$\mathcal{I}_T = O\left(H\left(d\ln(T^2/\delta) + dd_s + d^2\ln\left(1 + \|W^\star\|_2^2 TH/\sigma^2\right)\right)\ln\left(1 + \|W^\star\|_2^2 TH/\sigma^2\right)\right).$$

*Proof.* From the previous proposition, we know that $\sigma_t^2(s, a) = (\beta_t^2/\sigma^2)\|\phi(s, a)\|_{\Sigma_t^{-1}}^2$. Setting $\lambda = \sigma^2/\|W^\star\|_2^2$, we will have $\beta_t^2/\sigma^2 \geq 1$, which means that $\min\{\sigma_t^2(s, a), 1\} \leq (\beta_t^2/\sigma^2)\min\left\{\|\phi(s, a)\|_{\Sigma_t^{-1}}^2, 1\right\}$.

Note that $\beta_t$ is non-decreasing with respect to $t$, so $\beta_t \leq \beta_T$ for $T \geq t$, where

$$\beta_T = \sqrt{2\sigma^2 + 8\sigma^2(d_s \ln(5) + 2\ln(T^2/\delta) + \ln(4) + d\ln(1 + TH\|W^\star\|_2^2/\sigma^2))}$$

Also we have $\sum_{t=0}^{T-1} \sum_{h=0}^{H-1} \min\left\{ \|\phi(s_h^t, a_h^t)\|_{\Sigma_t^{-1}}^2, 1 \right\} \leq H \sum_{t=0}^{T-1} \min\left\{ \sum_{h=0}^{H-1} \|\phi(s_h^t, a_h^t)\|_{\Sigma_t^{-1}}^2, 1 \right\}$,
since $\min\{a_1, b_1\} + \min\{a_2, b_2\} \leq \min\{a_1 + a_2, b_1 + b_2\}$. Now call Lemma B.6 in [25], we have:

$$\sum_{t=0}^{T-1} \min\left\{ \sum_{h=0}^{H-1} \|\phi(s_h^t, a_h^t)\|_{\Sigma_t^{-1}}^2, 1 \right\} \leq 2\ln\left(\det(\Sigma_T)/\det(\lambda I)\right) = 2d\ln\left(1 + TH\|W^\star\|_2^2/\sigma^2\right).$$

$$(4)$$

Finally recall the definition of $\mathcal{I}_T$, we have:

$$\mathcal{I}_T = \sum_{t=0}^{T-1} \sum_{h=0}^{H-1} \min\left\{ \sigma_t^2(s_h^t, a_h^t), 1 \right\} \leq \frac{\beta_T^2}{\sigma^2} \sum_{t=0}^{T-1} \sum_{h=0}^{H-1} \min\left\{ \|\phi(s_h^t, a_h^t)\|_{\Sigma_t^{-1}}^2, 1 \right\} \leq \frac{\beta_T^2}{\sigma^2} 2Hd\ln(1 + \|W^\star\|_2^2 TH/\sigma^2)$$

$$\leq 2Hd\left(2 + 8\left(d_s \ln(5) + 2\ln(T^2/\delta) + \ln(4) + d\ln\left(1 + \|W^\star\|_2^2 TH/\sigma^2\right)\right)\right)\ln\left(1 + \|W^\star\|_2^2 TH/\sigma^2\right)$$

$$= H\left(4d + 32dd_s + 32d\ln(T^2/\delta) + 32d + 2d^2\ln\left(1 + \|W^\star\|_2^2 TH/\sigma^2\right)\right)\ln\left(1 + \|W^\star\|_2^2 TH/\sigma^2\right),$$

which concludes the proof. $\qquad\square$

**Extension to Infinite Dimensional RKHS** When $\phi : \mathcal{S} \times \mathcal{A} \mapsto \mathcal{H}$ where $\mathcal{H}$ is some infinite dimensional RKHS, we can bound our regret using the following intrinsic dimension:

$$\widetilde{d} = \max_{\{\{s_h^t, a_h^t\}_{h=0}^{H-1}\}_{t=0}^{T-1}} \ln\left(I + \frac{1}{\lambda} \sum_{t=0}^{T-1} \sum_{h=0}^{H-1} \phi(s_h^t, a_h^t)\phi(s_h^t, a_h^t)^\top\right).$$

In this case, recall Proposition 12, we have:

$$\beta_t \leq \beta_T \leq \sqrt{2\lambda\|W^\star\|_2^2 + 8\sigma^2\left(d_s \ln(5) + 2\ln(t^2/\delta) + \ln(4) + \ln\left(\det(\Sigma_T)/\det(\lambda I)\right)\right)}$$

$$\leq \sqrt{2\lambda\|W^\star\|_2^2 + 8\sigma^2\left(d_s \ln(5) + 2\ln(t^2/\delta) + \ln(4) + \widetilde{d}\right)}.$$

Also recall Eq. (4), we have:

$$\sum_{t=0}^{T-1} \min\left\{ \sum_{h=0}^{H-1} \|\phi(s_h^t, a_h^t)\|_{\Sigma_t^{-1}}^2, 1 \right\} \leq 2\ln\left(\det(\Sigma_T)/\det(\lambda I)\right) \leq 2\widetilde{d}.$$

Combine the above two, following similar derivation we had for finite dimensional setting, we have:

$$\mathcal{I}_T = \widetilde{O}\left(H\widetilde{d}^2 + H\widetilde{d}d_s\right).$$

### A.3 General Function Class $\mathcal{G}$ with Bounded Eluder dimension

**Proposition 14.** *Fix $\delta \in (0,1)$. Consider a general function class $\mathcal{G}$ where $\mathcal{G}$ is discrete, and $\sup_{g \in \mathcal{G}, s, a} \|g(s,a)\|_2 \leq G$. At iteration $t$, denote $\widehat{g}_t \in \arg\min_{g \in \mathcal{G}} \sum_{i=0}^{t-1} \sum_{h=0}^{H-1} \|g(s_h^i, a_h^i) - s_{h+1}^i\|_2^2$, and denote a version space $\mathcal{G}_t$ as:*

$$\mathcal{G}_t = \left\{ g \in \mathcal{G} : \sum_{i=0}^{t-1} \sum_{h=0}^{H-1} \left\|g(s_h^i, a_h^i) - \widehat{g}_t(s_h^i, a_h^i)\right\|_2^2 \leq c_t \right\}, \text{ with } c_t = 2\sigma^2 G^2 \ln(2t^2|\mathcal{G}|/\delta).$$

*The with probability at least $1 - \delta$, we have that for all $t$, and all $s, a$:*

$$\left\|\widehat{P}_t(\cdot|s,a) - P^\star(\cdot|s,a)\right\|_1 \leq \min\left\{ \frac{1}{\sigma} \max_{g_1 \in \mathcal{G}_t, g_2 \in \mathcal{G}_t} \|g_1(s,a) - g_2(s,a)\|_2, 2 \right\}.$$

*Proof.* Consider a fixed function $g \in \mathcal{G}$. Let us denote $z_h^t = \left\|g(s_h^t, a_h^t) - s_{h+1}^t\right\|_2^2 - \left\|g^\star(s_h^t, a_h^t) - s_{h+1}^t\right\|_2^2$. We have:

$$z_h^t = \left(g(s_h^t, a_h^t) - g^\star(s_h^t, a_h^t)\right)^\top \left(g(s_h^t, a_h^t) + g^\star(s_h^t, a_h^t) - 2g^\star(s_h^t, a_h^t) - 2\epsilon_h^t\right)$$
$$= \left\|g(s_h^t, a_h^t) - g^\star(s_h^t, a_h^t)\right\|_2^2 - 2(g(s_h^t, a_h^t) - g^\star(s_h^t, a_h^t))^\top \epsilon_h^t.$$

Since $\epsilon_h^t \sim \mathcal{N}(0, \sigma^2 I)$, we must have:

$$2(g(s_h^t, a_h^t) - g^\star(s_h^t, a_h^t))^\top \epsilon_h^t \sim \mathcal{N}(0, 4\sigma^2 \left\|g(s_h^t, a_h^t) - g^\star(s_h^t, a_h^t)\right\|_2^2)$$

Since $\sup_{g,s,a} \|g(s,a)\|_2 \le G$, then $2(g(s_h^t, a_h^t) - g^\star(s_h^t, a_h^t))^\top \epsilon_h^t$ is a $2\sigma G$ sub-Gaussian random variable.

Call Lemma 3 in [52], we have that with probability at least $1 - \delta$:

$$\sum_t \sum_h \left\|g(s_h^t, a_h^t) - s_{h+1}^t\right\|_2^2 \ge \sum_t \sum_h \left\|g^\star(s_h^t, a_h^t) - s_{h+1}^t\right\|_2^2 + 2\sum_t \sum_h \left\|g(s_h^t, a_h^t) - g^\star(s_h^t, a_h^t)\right\|_2^2 - 4\sigma^2 G^2 \ln(1/\delta).$$

Note that the above can also be derived directly using Azuma-Bernstein's inequality and the property of square loss. With a union bound over all $g \in \mathcal{G}$, we have that with probability at least $1 - \delta$, for all $g \in \mathcal{G}$.

$$\sum_t \sum_h \left\|g(s_h^t, a_h^t) - s_{h+1}^t\right\|_2^2 \ge \sum_t \sum_h \left\|g^\star(s_h^t, a_h^t) - s_{h+1}^t\right\|_2^2 + 2\sum_t \sum_h \left\|g(s_h^t, a_h^t) - g^\star(s_h^t, a_h^t)\right\|_2^2 - 4\sigma^2 G^2 \ln(|\mathcal{G}|/\delta).$$

Set $g = \widehat{g}_t$, and use the fact that $g_t$ is the minimizer of $\sum_t \sum_h \|g(s_h^t, a_h^t) - s_{h+1}^t\|_2^2$, we must have:

$$\sum_t \sum_h \left\|\widehat{g}_t(s_h^t, a_h^t) - g^\star(s_h^t, a_h^t)\right\|_2^2 \le 2\sigma^2 G^2 \ln(2t^2|\mathcal{G}|/\delta).$$

Namely we prove that our version space $\mathcal{G}_t$ contains $g^\star$ for all $t$. Thus, we have:

$$\left\|\widehat{P}_t(\cdot|s,a) - P^\star(\cdot|s,a)\right\|_1 \le \frac{1}{\sigma} \|\widehat{g}_t(s,a) - g^\star(s,a)\|_2 \le \frac{1}{\sigma} \sup_{g_1 \in \mathcal{G}_t, g_2 \in \mathcal{G}_t} \|g_1(s,a) - g_2(s,a)\|_2,$$

where the last inequality holds since both $g^\star$ and $\widehat{g}_t$ belong to the version $\mathcal{G}_t$.

$\square$

Now we bound the information gain $\mathcal{I}_T$ below. The proof mainly follows from the proof in [42].

**Lemma 15** (Lemma 1 in [42]). *Denote* $\beta_t = 2\sigma^2 G^2 \ln(t^2|\mathcal{G}|/\delta)$. *Let us denote the uncertainty measure* $w_{t;h} = \sup_{f_1, f_2 \in \mathcal{G}_t} \|f_1(s_h^t, a_h^t) - f_2(s_h^t, a_h^t)\|_2$ *(note that* $w_{t;h}$ *is non-negative). We have:*

$$\sum_{i=0}^{t-1} \sum_{h=0}^{H-1} \mathbf{1}\{w_{t;h}^2 > \epsilon\} \le \left(\frac{4\beta_t}{\epsilon} + H\right) d_E(\sqrt{\epsilon}).$$

**Proposition 16** (Bounding $\mathcal{I}_T$). *Denote* $d = d_E(1/TH)$. *We have*

$$\mathcal{I}_T = \left(1/\sigma^2 + HdG^2/\sigma^2 + 8G^2 \ln(T^2|\mathcal{G}|/\delta)d\ln(TH)\right).$$

*Proof.* Note that the uncertainty measures $w_{t;h}$ are non-negative. Let us reorder the sequence and denote the ordered one as $w_1 \ge w_2 \ge w_3 \cdots \ge w_{TH-H}$. For notational simplicity, denote $M = TH - H$ We have:

$$\sum_{i=0}^{T-1} \sum_{h=0}^{H-1} w_{t;h}^2 = \sum_{i=0}^{M-1} w_i^2 \le 1 + \sum_i w_i^2 \mathbf{1}\{w_i^2 \ge \frac{1}{M}\},$$

where the last inequality comes from the fact that $\sum_i w_i^2 \mathbf{1}\{w_i^2 < 1/M\} \le M\frac{1}{M} = 1$. Consider any $w_t$ where $w_t^2 \ge 1/M$. In this case, we know that $w_1^2 \ge w_2^2 \ge \cdots \ge w_t^2 \ge 1/M$. This means that:

$$t \le \sum_i \sum_h \mathbf{1}\{w_{t;h}^2 > w_t^2\} \le \left(\frac{4\beta_T}{w_t^2} + H\right) d_E(\sqrt{w_t}) \le \left(\frac{4\beta_T}{w_t^2} + H\right) d_E(1/M),$$

where the second inequality uses the lemma above, and the last inequality uses the fact that $d_E(\epsilon)$ is non-decreasing when $\epsilon$ gets smaller. Denote $d = d_E(1/M)$. The above inequality indicates that $w_t^2 \le \frac{4\beta_T d}{t - Hd}$. This means that for any $w_t^2 \ge 1/M$, we must have $w_t^2 \le 4\beta_T d/(t - Hd)$. Thus, we have:

$$\sum_{i=0}^{T-1} \sum_{h=0}^{H-1} w_{t;h}^2 \le 1 + HdG^2 + \sum_{\tau=Hd+1}^{M} w_\tau^2 \mathbf{1}\{w_\tau^2 \ge 1/M\} \le 1 + HdG^2 + 4\beta_T d \ln(M)$$
$$= 1 + HdG^2 + 4\beta_T d \ln(TH).$$

Finally, recall the definition of $\mathcal{I}_T$, we have:

$$\sum_{t=0}^{T-1} \sum_{h=0}^{H-1} \min\{\sigma_t^2(s_h^t, a_h^t), 1\} \le \sum_{t=0}^{T-1} \sum_{h=0}^{H-1} \sigma_t^2(s_h^t, a_h^t) \le \frac{1}{\sigma^2} \sum_{t=0}^{T-1} \sum_{h=0}^{H-1} w_{t;h}^2 \le \frac{1}{\sigma^2} \left( 1 + HdG^2 + 4\beta_T d \ln(TH) \right).$$

This concludes the proof. $\qquad\square$

### A.4  Proof of Theorem 7

This section provides the proof of Theorem 7.

First we present the reduction from a bandit optimization problem to ILFO.

Consider a Multi-armed bandit (MAB) problem with $A$ many actions $\{a_i\}_{i=1}^A$. Each action's ground truth reward $r_i$ is sampled from a Gaussian with mean $\mu_i$ and variance 1. Without loss of generality, assume $a_1$ is the optimal arm, i.e., $\mu_1 \ge \mu_i \,\forall\, i \ne 1$. We convert this MAB instance into an MDP. Specifically, set $H = 2$. Suppose we have a fixed initial state $s_0$ which has $A$ many actions. For the one step transition, we have $P(\cdot|s_0, a_i) = \mathcal{N}(\mu_i, 1)$, i.e., $g^*(s_0, a_i) = \mu_i$. Here we denote the optimal expert policy $\pi^e$ as $\pi^e(s_0) = a_1$, i.e., expert policy picks the optimal arm in the MAB instance. Hence, when executing $\pi^e$, we note that the state $s_1$ generated from $\pi^e$ is simply the stochastic reward of $a_1$ in the original MAB instance. Assume that we have observed infinitely many such $s_1$ from the expert policy $\pi^e$, i.e., we have infinitely many samples of expert state data, i.e., $N \to \infty$. Note, however, we do not have the actions taken by the expert (since this is the ILFO setting). This expert data is equivalent to revealing the optimal arm's mean reward $\mu_1$ to the MAB learner a priori. Hence solving the ILFO problem on this MDP is no easier than solving the original MAB instance with additional information which is that optimal arm's mean reward is $\mu_1$ (but the best arm's identity is unknown).

Below we show the lower bound for solving the MAB problem where the optimal arm's mean is known.

**Theorem 17.** *Consider best arm identification of Gaussian MAB with the additional information that the optimal arm's mean reward is $\mu$. For any algorithm, there exists a MAB instance with number of arms $A \ge 2$, such that the expected cumulative regret is still $\Omega(\sqrt{AT})$, i.e., the additional information does not help improving the worst-case regret bound to solve the MAB instance.*

*Proof of Theorem 17.* Below, we will construct $A$ many MAB instances where each instance has $A$ many arms and each arm has a Gaussian reward distribution with the fixed variance $\sigma^2$. Each of the $A$ instances has the maximum mean reward equal to $\Delta$, i.e., all these $A$ instances have the same maximum arm mean reward. Consider any algorithm Alg that maps $\Delta$ together with the history of the interactions $\mathcal{H}_t = \{a_0, r_0, a_1, r_1, \ldots, a_{t-1}, r_{t-1}\}$ to a distribution over $A$ actions. We will show for any such algorithm alg that knows $\Delta$, with constant probability, there must exist a MAB instance from the $A$ many MAB instances, such that Alg suffers at least $\Omega(\sqrt{AT})$ regret where $T$ is the number of iterations.

Now we construct the $A$ instances as follows. Consider the $i$-th instance ($i = 1, \ldots, A$). For arm $j$ in the i-th instance, we define its mean as $\mu_j^i = \mathbf{1}\{i = j\}\Delta$. Namely, for MAB instance $i$, its arms have mean reward zero everywhere except that the $i$-th arm has reward mean $\Delta$. Note that all these MAB instances have the same maximum mean reward, i.e., $\Delta$. Hence, we cannot distinguish them by just revealing $\Delta$ to the learner.

We will construct an additional MAB instance (we name it as $0$-th MAB instance) whose arms have reward mean zero. Note that this MAB instance has maximum mean reward 0 which is different from

the previous $A$ MAB instances that we constructed. However, we will only look at the regret of Alg on the previously constructed $A$ MAB instances. I.e., we do not care about the regret of $\text{Alg}(\Delta, \mathcal{H}_t)$ on the 0-th MAB instance.

Let us denote $\mathbb{P}_i$ (for $i = 0, \ldots, A$) as the distribution of the outcomes of algorithm $\text{Alg}(\Delta, \mathcal{H}_t)$ interacting with MAB instance $i$ for $n$ iterations, and $\mathbb{E}_j[N_i(T)]$ as the expected number of times arm $i$ is pulled by $\text{Alg}(\Delta, \mathcal{H}_t)$ in MAB instance $j$. Consider MAB instance $i$ with $i \geq 1$:

$$\mathbb{E}_i[N_i(T)] - \mathbb{E}_0[N_i(T)] \leq T \, \|\mathbb{P}_i - \mathbb{P}_0\|_1 \leq T \sqrt{\text{KL}(\mathbb{P}_0, \mathbb{P}_i)} \leq T \sqrt{\Delta^2 \mathbb{E}_0[N_i(T)]},$$

where the last step uses the fact that we are running the same algorithm $\text{Alg}(\Delta, \mathcal{H}_t)$ on both instance 0 and instance $i$ (i.e., same policy for generating actions), and thus, $\text{KL}(\mathbb{P}_0, \mathbb{P}_i) = \sum_{j=1}^{A} \mathbb{E}_0[N_j(T)]\text{KL}\left(q_0(j), q_i(j)\right)$ (Lemma 15.1 in [35]), where $q_i(j)$ is the reward distribution of arm $j$ at instance $i$. Also recall that for instance 0 and instance $i$, their rewards only differ at arm $i$.

This implies that:

$$\mathbb{E}_i[N_i(T)] \leq \mathbb{E}_0[N_i(T)] + T\sqrt{\Delta^2 \mathbb{E}_0[N_i(T)]}.$$

Sum over $i = 1, \ldots, A$ on both sides, we have:

$$\sum_{i=1}^{A} \mathbb{E}_i[N_i(T)] \leq T + T \sum_{i=1}^{A} \sqrt{\Delta^2 \mathbb{E}_0[N_i(T)]} \leq T + T\sqrt{A}\sqrt{\sum_{i=1}^{A} \Delta^2 \mathbb{E}_0[N_i(T)]}$$

$$\leq T + T\sqrt{A}\sqrt{\Delta^2 T}$$

Now let us calculate the regret of $\text{Alg}(\Delta, \mathcal{H}_t)$ on $i$-th instance, we have:

$$R_i = T\Delta - \mathbb{E}_i[N_i(T)]\Delta.$$

Sum over $i = 1, \ldots, A$, we have:

$$\sum_{i=1}^{A} R_i = \Delta \left( AT - \sum_{i=1}^{A} \mathbb{E}_i[N_i(T)] \right) \geq \Delta \left( AT - T - T\sqrt{A\Delta^2 T} \right)$$

Set $\Delta = c\sqrt{A/T}$ for some $c$ that we will specify later, we get:

$$\sum_{i=1}^{A} R_i \geq c\sqrt{\frac{A}{T}} \left( AT - T - cAT \right).$$

Set $c = 1/4$, we get:

$$\sum_{i=1}^{A} R_i \geq c\sqrt{\frac{A}{T}} \left( AT - T - cAT \right) \geq \frac{1}{4}\sqrt{AT} \left( A - 1 - A/4 \right) = \frac{1}{4}\sqrt{AT} \left( 3A/4 - 1 \right) \geq \frac{1}{4}\sqrt{AT} \left( A/4 \right),$$

assuming $A \geq 2$.

Thus there must exist $i \in \{1, \ldots, A\}$, such that:

$$R_i \geq \frac{1}{16}\sqrt{AT}.$$

Note that the above construction considered any algorithm $\text{Alg}(\Delta, \mathcal{H}_t)$ that maps $\Delta$ and history to action distributions. Thus it concludes the proof. $\qquad\square$

The hardness result in Theorem 17 and the reduction from MAB to ILFO together implies the lower bound for ILFO in Theorem 7, namely solving ILFO with cumulative regret smaller then $O(\sqrt{AT})$ will contradict the MAB lower bound in Theorem 17.

# B Auxiliary Lemmas

**Lemma 18** (Simulation Lemma). *Consider any two functions $f : \mathcal{S} \times \mathcal{A} \mapsto [0,1]$ and $\widehat{f} : \mathcal{S} \times \mathcal{A} \mapsto [0,1]$, any two transitions $P$ and $\widehat{P}$, and any policy $\pi : \mathcal{S} \mapsto \Delta(\mathcal{A})$. We have:*

$$
V_{P;f}^{\pi} - V_{\widehat{P},\widehat{f}}^{\pi} = \sum_{h=0}^{H-1} \mathbb{E}_{s,a \sim d_P^{\pi}} \left[ f(s,a) - \widehat{f}(s,a) + \mathbb{E}_{s' \sim P(\cdot|s,a)} V_{\widehat{P},\widehat{f};h}^{\pi}(s') - \mathbb{E}_{s' \sim \widehat{P}(\cdot|s,a)} V_{\widehat{P},\widehat{f};h}^{\pi}(s') \right]
$$

$$
\leq \sum_{h=0}^{H-1} \mathbb{E}_{s,a \sim d_P^{\pi}} \left[ f(s,a) - \widehat{f}(s,a) + \|V_{\widehat{P},\widehat{f};h}^{\pi}\|_{\infty} \|P(\cdot|s,a) - \widehat{P}(\cdot|s,a)\|_1 \right].
$$

*where $V_{P,f;h}^{\pi}$ denotes the value function at time step $h$, under $\pi, P, f$.*

Such simulation lemma is standard in model-based RL literature and can be found, for instance, in the proof of Lemma 10 from [58].

**Lemma 19.** *Consider two Gaussian distribution $P_1 := \mathcal{N}(\mu_1, \sigma^2 I)$ and $P_2 := \mathcal{N}(\mu_2, \sigma^2 I)$. We have:*

$$
\|P_1 - P_2\|_1 \leq \frac{1}{\sigma} \|\mu_1 - \mu_2\|_2.
$$

The above lemma can be proved by Pinsker's inequality and the closed-form of the KL divergence between $P_1$ and $P_2$.

# C Implementation Details

## C.1 Environment Setup and Benchmarks

This section sketches the details of how we setup the environments. We utilize the standard environment horizon of $500, 50, 200$ for `Cartpole-v1`, `Reacher-v2`, `Cartpole-v0`. For `Swimmer-v2`, `Hopper-v2` and `Walker2d-v2`, we work with the environment horizon set to $400$ [33, 39, 38, 48, 32]. Furthermore, for `Hopper-v2`, `Walker2d-v2`, we add the velocity of the center of mass to the state parameterization [48, 38, 32]. As noted in the main text, the expert policy is trained using NPG/TRPO [26, 54] until it hits a value of (approximately) $460, -10, 38, 3000, 2000, 170$ for `Cartpole-v1`, `Reacher-v2`, `Swimmer-v2`, `Hopper-v2`, `Walker2d-v2`, `Cartpole-v0` respectively. Furthermore, for `Walker2d-v2` we utilized pairs of states $(s, s')$ for defining the feature representation used for parameterizing the discriminator. All the results presented in the experiments section are averaged over five seeds. Furthermore, in terms of baselines, we compare `MobILE` to BC, BC-O, ILPO, GAIL and GAIFO. Note that BC/GAIL has access to expert actions whereas our algorithm does not have access to the expert actions. We report the average of the best performance offered by BC/BC-O when run with five seeds, even if this occurs at different epochs for each of the runs - this gives an upper hand to BC/BC-O. Moreover, note that for BC, we run the supervised learning algorithm for $500$ passes. Furthermore, we run BC-O/GAIL with same number of online samples as `MobILE` in order to present our results. Furthermore, we used 2 CPUs with 16-32 GB of RAM usage to perform all our benchmarking runs implemented in Pytorch. Finally, our codebase utilizes Open-AI's implementation of TRPO [15] for environments with discrete actions, and the MJRL repository [47] for working with continuous action environments. With regards to results in the main paper, our bar graph presenting normalized results was obtained by dividing every algorithm's performance (mean/standard deviation) by the expert mean; for `Reacher-v2` because the rewards themselves are negative, we first added a constant offset to make all the algorithm's performance to become positive, then, divided by the mean of expert policy.

## C.2 Practical Implementation of `MobILE`

We will begin with presenting the implementation details of `MobILE` (refer to Algorithm 2):

---

**Algorithm 2** `MobILE`: Model-based Imitation Learning and Exploring for ILFO (used in practical implementation)

---

1: **Require**: Expert Dataset $\mathcal{D}_e$, Access to dynamics of the true environment i.e. $P^\star$.
2: Initialize Policy $\pi_0$, Discriminator $w_0$, Replay Buffer of pre-determined size $\mathcal{D}$, Dynamics Model $\widehat{P}_{-1}$, Bonus $b_{-1}$.
3: **for** $t = 0, \cdots, T-1$ **do**
4:     **Online Interaction**: Execute $\pi_t$ in true environment $P^\star$ to get samples $\mathcal{S}_t$.
5:     **Update replay buffer**: $\mathcal{D} = \text{Replay-Buffer-Update}(\mathcal{D}, \mathcal{S}_t)$ (refer to section C.2.2).
6:     **Update dynamics model**: Obtain $\widehat{P}_t$ by starting at $\widehat{P}_{t-1}$ and update using replay buffer $\mathcal{D}$ (refer to section C.2.1).
7:     **Bonus Update**: Update bonus $b_t : \mathcal{S} \times \mathcal{A} \to \mathbb{R}^+$ using replay buffer $\mathcal{D}$ (refer to section C.2.3).
8:     **Discriminator Update**: Update discriminator as $w_t \leftarrow \arg\max_w L(w; \pi_t, \widehat{P}_t, b_t, \mathcal{D}_e)$ (refer to section C.2.4).
9:     **Policy Update**: Perform incremental policy update through approximate minimization of $L(\cdot)$,
                i.e.: $\pi_t \leftarrow \arg\min_\pi L(\pi; w_t, \widehat{P}_t, b_t, \mathcal{D}_e)$ by running $K_{PG}$ steps of TRPO
    (refer to section C.2.5).
10: **end for**
11: **Return** $\pi_T$.

---

### C.2.1 Dynamics Model Training

As detailed in the main paper, we utilize a class of Gaussian Dynamics Models parameterized by an MLP [48], i.e. $\widehat{P}(s, a) := \mathcal{N}(h_\theta(s, a), \sigma^2 I)$, where, $h_\theta(s, a) = s + \sigma_{\Delta_s} \cdot \text{MLP}_\theta(s_c, a_c)$, where, $\theta$ are MLP's trainable parameters, $s_c = (s - \mu_s)/\sigma_s$, $a_c = (a - \mu_a)/\sigma_a$ with $\mu_s, \mu_a$ (and $\sigma_s, \sigma_a$) being the mean of states, actions (and standard deviation of states and actions) in the replay buffer $\mathcal{D}$. Note that we predict normalized state differences instead of the next state directly.

In practice, we fine tune our estimate of dynamics models based on the new contents of the replay buffer as opposed to re-training the models from scratch, which is computationally more expensive. In particular, we start from the estimate $\widehat{P}_{t-1}$ in the $t-1$ epoch and perform multiple updates gradient updates using the contents of the replay buffer $\mathcal{D}$. We utilize constant stepsize SGD with momentum [60] for updating our dynamics models. Since the distribution of $(s, a, s')$ pairs continually drift as the algorithm progresses (for instance, because we observe a new state), we utilize gradient clipping to ensure our model does not diverge due to the aggressive nature of our updates.

### C.2.2 Replay Buffer

Since we perform incremental training of our dynamics model, we utilize a replay buffer of a fixed size rather than training our dynamics model on all previously collected online $(s, a, s')$ samples. Note that the replay buffer could contain data from all prior online interactions should we re-train our dynamics model from scratch at every epoch.

### C.2.3 Design of Bonus Function

We utilize an ensemble of two transition dynamics models incrementally learned using the contents of the replay buffer. Specifically, given the models $h_{\theta_1}(\cdot)$ and $h_{\theta_2}(\cdot)$, we compute the discrepancy as: $\delta(s, a) = ||h_{\theta_1}(s, a) - h_{\theta_2}(s, a)||_2$. Moreover, given a replay buffer $\mathcal{D}$, we compute the maximum discrepancy as $\delta_{\mathcal{D}} = \max_{(s,a,s')\sim\mathcal{D}} \delta(s, a)$. We then set the bonus as $b(s, a) = \min(1, \delta(s, a)/\delta_{\mathcal{D}}) \cdot \lambda$, thus ensuring the magnitude of our bonus remains bounded between $[0, \lambda]$ roughly.

### C.2.4 Discriminator Update

Recall that $f_w(s) = w^\top \psi(s)$, where $w$ are the parameters of the discriminator. Given a policy $\pi$, the update for the parameters $w$ take the following form:

$$\max_{w: ||w||_2^2 \leq \zeta} L(w; \pi, \widehat{P}, b, \mathcal{D}_e) := \mathbb{E}_{(s,a)\sim d_{\widehat{P}}^\pi}[f_w(s) - b(s, a)] - \mathbb{E}_{s\sim\mathcal{D}_e}[f_w(s)]$$

$$\equiv \max_{w} L_{\zeta}(w; \pi, \widehat{P}, b, \mathcal{D}_e) = \mathbb{E}_{(s,a) \sim d_{\widehat{P}}^{\pi}} [f_w(s) - b(s,a)] - \mathbb{E}_{s \sim \mathcal{D}_e} [f_w(s)] - \frac{1}{2} \cdot \left( \|w\|_2^2 - \zeta \right),$$

$$\implies \partial_w L_{\zeta}(w; \pi, \widehat{P}, b, \mathcal{D}_e) = \mathbb{E}_{s \sim d_{\widehat{P}}^{\pi}} [\psi(s)] - \mathbb{E}_{s \sim \mathcal{D}_e} [\psi(s)] - w \in 0,$$

where, $\partial_w L_{\zeta}(w; \pi, \widehat{P}, b, \mathcal{D}_e)$ denotes the sub-differential of $L_{\zeta}(\cdot)$ wrt $w$. This in particular implies the following:

1. **Exact Update:** $w^* = \mathcal{P}_{\mathcal{B}(\zeta)} \left( \mathbb{E}_{s \sim d_{\widehat{P}}^{\pi}} [\psi(s)] - \mathbb{E}_{s \sim \mathcal{D}_e} [\psi(s)] \right)$, $\mathcal{P}.$ is the projection operator, and $\mathcal{B}(\zeta)$ is the $\zeta$−norm ball.

2. **Gradient Ascent Update:** $w_{t+1} = \mathcal{P}_{\mathcal{B}(\zeta)} \left( (1 - \eta_w) w_t + \eta_w \cdot \left( \mathbb{E}_{s \sim d_{\widehat{P}}^{\pi}} [\psi(s)] - \mathbb{E}_{s \sim \mathcal{D}_e} [\psi(s)] \right) \right)$, $\eta_w > 0$ is the step-size.

We found empirically either of the updates to work reasonably well. In the `Swimmer-v2` task, we use the gradient ascent update with $\eta_w = 0.67$, and, in the other tasks, we utilize the exact update. Furthermore, we empirically observe the gradient ascent update to yield more stability compared to the exact updates. In the case of `Walker2d-v2`, we found it useful to parameterize the discriminator based on pairs of states $(s, s')$.

### C.2.5 Model-Based Policy Update

Once the maximization of the discriminator parameters $w$ is performed, consider the policy optimization problem, i.e.,

$$\min_{\pi} L(\pi; w, \widehat{P}, b, \mathcal{D}_e) := \mathbb{E}_{(s,a) \sim d_{\widehat{P}}^{\pi}} [f_w(s) - b(s,a)] - \mathbb{E}_{s \sim \mathcal{D}_e} [f_w(s)]$$

$$\equiv \min_{\pi} L(\pi; w, \widehat{P}, b, \mathcal{D}_e) = \mathbb{E}_{(s,a) \sim d_{\widehat{P}}^{\pi}} [f_w(s) - b(s,a)]$$

Hence we perform model-based policy optimization under $\widehat{P}$ and cost function $f_w(s) - b(s,a)$. In practice, we perform approximate minimization of $L(\cdot)$ by incrementally updating the policy using $K_{PG}$-steps of policy gradient, where, $K_{PG}$ is a tunable hyper-parameter. In our experiments, we find that setting $K_{PG}$ to be around 10 to generally be a reasonable choice (for precise values, refer to Table 2). This paper utilizes TRPO [54] as our choice of policy gradient method; note that this can be replaced by other alternatives including PPO [55], SAC [20] *etc.* Similar to practical implementations of existing policy gradient methods, we implement a reward filter by clipping the IPM reward $f(s)$ by truncating it between $c_{\min}$ and $c_{\max}$ as this leads to stability of the policy gradient updates. Note that the minimization is done with access to $\widehat{P}$, which implies we perform *model-based* planning. Empirically, for purposes of tuning the exploration-imitation parameter $\lambda$, we minimize a surrogate namely: $\mathbb{E}_{(s,a) \sim d_{\widehat{P}}^{\pi}} [(1 - \lambda) \cdot f_w(s) - b(s,a)]$ (recall that $b(s,a)$ has a factor of $\lambda$ associated with it). This ensures that we can precisely control the magnitude of the bonuses against the IPM costs, which, in our experience is empirically easier to work with.

### C.3 Hyper-parameter Details

This section presents an overview of the list of hyper-parameters necessary to implement Algorithm 1 in practice, as described in Algorithm 2. The list of hyper-parameters is precisely listed out in Table 2. The hyper-parameters are broadly categorized into ones corresponding to various components of `MobILE`, namely, (a) environment specifications, (b) dynamics model, (c) ensemble based bonus, (d) IPM parameterization, (e) Policy parameterization, (f) Planning algorithm parameters, (g) Critic parameterization. Note that if there a hyper-parameter that has not been listed, for instance, say, the value of momentum for the ADAM optimizer in the critic, this has been left as is the default value defined in Pytorch.

## D  Additional Experimental Results

### D.1  Modified `Cartpole-v0` environment with noise added to transition dynamics

| Parameter | Cartpole-v1 | Reacher-v2 | Swimmer-v2 | Cartpole-v0 | Hopper-v2 | Walker2d-v2 |
|---|---|---|---|---|---|---|
| **Environment Specifications** | | | | | | |
| Horizon $H$ | 500 | 50 | 400 | 200 | 400 | 400 |
| Expert Performance ($\approx$) | 460 | $-10$ | 38 | 181 | 3000 | 2000 |
| # online samples per outer loop | $2 \cdot H$ | $2 \cdot H$ | $2 \cdot H$ | $2 \cdot H$ | $8 \cdot H$ | $3 \cdot H$ |
| **Dynamics Model** | | | | | | |
| Architecture/Non-linearity | MLP(64, 64)/ReLU | MLP(64, 64)/ReLU | MLP(512, 512)/ReLU | MLP(64, 64)/ReLU | MLP(512, 512)/ReLU | MLP(512, 512)/ReLU |
| Optimizer(LR, Momentum, Batch Size) | SGD(0.005, 0.99, 256) | SGD(0.005, 0.99, 256) | SGD(0.005, 0.99, 256) | SGD(0.005, 0.99, 256) | SGD(0.005, 0.99, 256) | SGD(0.005, 0.99, 256) |
| # train passes per outer loop | 20 | 100 | 100 | 20 | 50 | 200 |
| Grad Clipping | 2.0 | 2.0 | 1.0 | 2.0 | 4.0 | 1.0 |
| Replay Buffer Size | $10 \cdot H$ | $10 \cdot H$ | $10 \cdot H$ | $10 \cdot H$ | $16 \cdot H$ | $15 \cdot H$ |
| **Ensemble based bonus** | | | | | | |
| # models/bonus range | 2/[0, 1] | 2/[0, 1] | 2/[0, 1] | 2/[0, 1] | 2/[0, 1] | 2/[0, 1] |
| **IPM parameters** | | | | | | |
| Step size for $w$ update ($\eta_w$) | Exact | Exact | 0.33 | Exact | Exact | Exact |
| # RFFs/BW Heuristic | 128/0.1 quantile | 128 / 0.1 quantile | 128 / 0.1 quantile | 128 / 0.1 quantile | 128 / 0.1 quantile | 128 / 0.1 quantile |
| **Policy parameterization** | | | | | | |
| Architecture/Non-linearity | MLP(64, 64)/TanH | MLP(64, 64)/TanH | MLP(64, 64)/TanH | MLP(32, 32)/TanH | MLP(32, 32)/TanH | MLP(32, 32)/TanH |
| Policy Constraints | None | None | None | None | $\log \sigma_{\min} = -1.0$ | $\log \sigma_{\min} = -2.0$ |
| **Planning Algorithm** | | | | | | |
| # model samples per TRPO step | $2 \cdot H$ | $10 \cdot H$ | $4 \cdot H$ | $4 \cdot H$ | $8 \cdot H$ | $20 \cdot H$ |
| # TRPO steps per outer loop ($K_{PG}$) | 3 | 10 | 20 | 5 | 10 | 15 |
| TRPO Parameters (CG iters, dampening, kl, gae$_\lambda$, $\gamma$) | (50, 0.001, 0.01, 0.97, 0.995) | (100, 0.001, 0.01, 0.97, 0.995) | (100, 0.001, 0.01, 0.97, 0.995) | (100, 0.001, 0.01, 0.97, 0.995) | (10, 0.0001, 0.025, 0.97, 0.995) | (10, 0.0001, 0.025, 0.97, 0.995) |
| **Critic parameterization** | | | | | | |
| Architecture/Non-linearity | MLP(128, 128)/ReLU | MLP(128, 128)/ReLU | MLP(128, 128)/ReLU | MLP(32, 32)/ReLU | MLP(128, 128)/ReLU | MLP(128, 128)/ReLU |
| Optimizer (LR, Batch Size, $\epsilon$, Regularization) | Adam(0.001, 64, $1e-5$, 0) | Adam(0.001, 64, $1e-5$, 0) | Adam(0.001, 64, $1e-5$, 0) | Adam(0.001, 64, $1e-5$, 0) | Adam(0.001, 64, $1e-8$, $1e-3$) | Adam(0.001, 64, $1e-8$, $1e-3$) |
| # train passes per TRPO update | 1 | 1 | 1 | 1 | 2 | 2 |

Table 2: List of various Hyper-parameters employed in `MobILE`'s implementation.

We consider a stochastic variant of `Cartpole-v0`, wherein, we add additive Gaussian noise of variance unknown to the learner in order to make the transition dynamics of the environment to be stochastic. Specifically, we train an expert of value $\approx$ 170 in `Cartpole-v0` with stochastic dynamics using TRPO. Now, using 20 trajectories drawn from this expert, we wish to consider solving the ILFO problem using `MobILE` as well as other baselines including BC, BC-O, ILPO, GAIL and GAIFO. Figure 4 presents the result of this comparison. Note that `MobILE` compares favorably against other baseline methods - in particular, BC tends suffer in environments like `Cartpole-v0` with stochastic dynamics because of increased generalization error of the supervised learning algorithm used for learning a policy. Our algorithm is competitive with both BC-O, GAIL, GAIFO and ILPO. Note that BC-O tends to outperform BC both in `Cartpole-v1` and in `Cartpole-v0` (with stochastic dynamics).

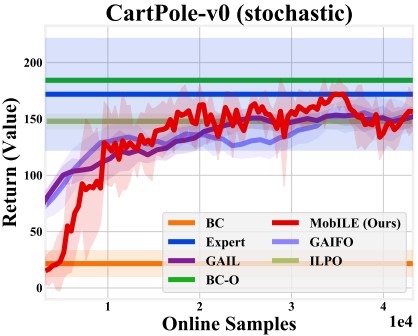

Figure 4: Learning curves for `Cartpole-v0` with stochastic dynamics with 20 expert trajectories comparing `MobILE` with BC, BC-O, GAIL, GAIFO and ILPO.

### D.2 Swimmer Learning Curves

We supplement the learning curves for `Swimmer-v2` (with 40 expert trajectories) with the learning curves for `Swimmer-v2` with 10 expert trajectories in figure 5. As can be seen, `MobILE` outperforms baseline algorithms such as BC, BC-O, ILPO, GAIL and GAIFO in `Swimmer-v2` with both 40 and 10 expert trajectories. The caveat is that for 10 expert trajectories, all algorithms tend to show a lot more variance in their behavior and this reduces as we move to the 40 expert trajectory case.

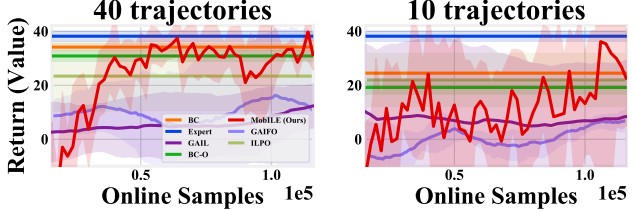

Figure 5: Learning curves for `Swimmer-v2` with 40 (left) and 10 (right) expert trajectories comparing `MobILE` with BC, BC-O, ILPO, GAIL and GAIFO. `MobILE` continues to perform well relative to all other benchmarks with both 10 and 40 expert trajectories. The variance of the algorithm as well as the benchmarks is notably higher with lesser number of expert trajectories.

## D.3 Additional Results

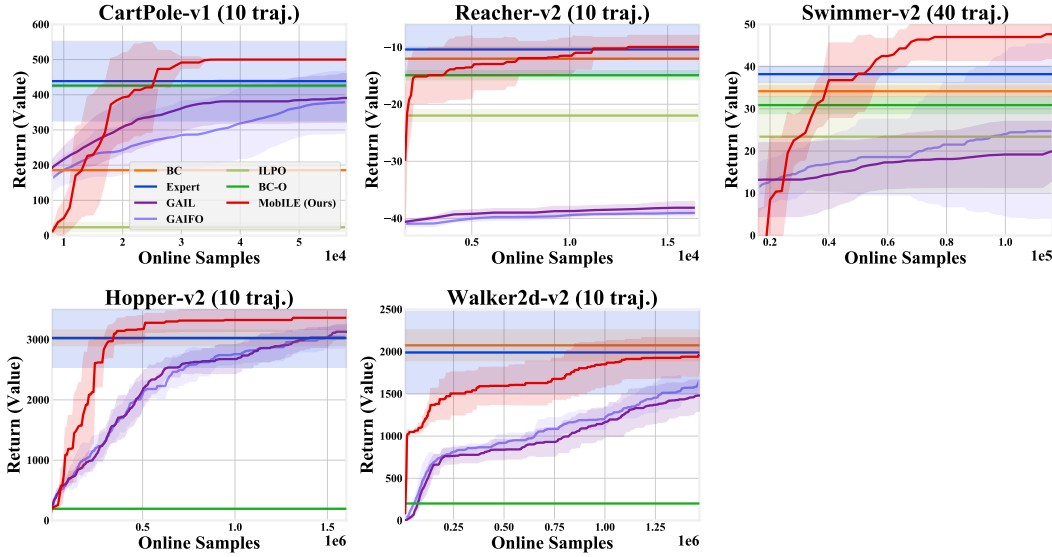

Figure 6: Learning curves tracking the running maximum averaged across seeds comparing `MobILE` against BC, BC-O, ILPO, GAIL and GAIFO. `MobILE` tends to reach expert performance consistently and in a more sample efficient manner.

In this section, we give another view of our results for `MobILE` compared against the baselines (BC/BC-O/ILPO/GAIL/GAIFO) by tracking the running maximum of each policy's value averaged across seeds. Specifically, for every iteration $t$, we plot the best policy performance obtained by the algorithm so far averaged across seeds (note that this quantity is monotonic, since the best policy obtained so far can never be worse at a later point of time when running the algorithm). For BC/BC-O/ILPO, we present a simplified view by picking the best policy obtained through the course of running the algorithm and averaging it across seeds (so the curves are flat lines). As figure 6 shows, `MobILE` reliably hits expert performance faster than GAIL and GAIFO while often matching/outperforming ILPO/BC/BC-O.

## D.4 Ablation Study on Number of Models used for Strategic Exploration Bonus

In this experiment, we present an ablation study on using more number of models in the ensemble for setting the strategic exploration bonus. Figure 7 suggests that even utilizing two models for purposes of setting the bonus is effective from a practical perspective.

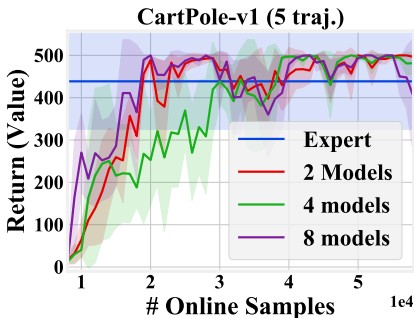

Figure 7: Learning curves for `Cartpole-v1` with varying number of dynamics models for assigning bonuses for strategic exploration.