# OpenReview forum: "MobILE: Model-Based Imitation Learning From Observation Alone"
_NeurIPS.cc/2021/Conference — NeurIPS 2021 Poster_

### Official Review · Reviewer_eB6p · 2021-07-12

**Rating:** 6
**Confidence:** 3

**Summary:**

This paper investigates the problem of imitation learning from observation alone (ILFO). The authors leverage the idea of optimism in the face of uncertainty and propose a novel model-based framework for ILFO that can handle both discrete and continuous settings. The authors provide a regret bound of the framework and show the exponential gap between IL and ILFO and the benefit of uncertainty-based exploration. The authors further corroborate the theoretical and algorithmic results with experiments in simulation.

**Limitations And Societal Impact:**

yes

**Main Review:**

Strengths:

Although the idea of optimism in the face of uncertainty is not new, its application in ILFO seems noval and inspiring. The framework that the authors develop is general enough to work under both discrete and continuous settings, and it does not require the existence of an inverse dynamics model which is required by some of the previous works. In terms of theoretical findings, the authors show an exponential gap between IL and ILFO in the regret bound with respect to the number of actions, which is interesting. In addition, the authors theoretically justify the benefits of the confidence-based exploration. Furthermore, the experimental in simulation results look encouraging.

Comments and questions:

I appreciated that the authors are explicit about the realizability assumption on cost and transition functions. However it may be strong in practical applications. Can the current analysis be extended to lift the assumption in a straightforward way?

In Theorem 3, the rate applies to the best policy among the T polices. However, in Algorithm 1 the last policy is returned. Is it guaranteed that Algorithm 1 will always converge?

How’s the dependence on horizon H of the proposed algorithm compared to other existing ILFO algorithms including FAIL?
In Line 6 of Algorithm, does it mean that N expert states are required in each of the T iterations?

The paper is easy to follow in general. However, I think there’s still room for improvement.
- I did not find where regret (mentioned in e.g. Page 2 Line 47) is formally defined or motivated in the paper before it appears implicitly late in the paper -- Theorem 3.
- I don’t quite get the definition of Information Gain in Eq. (3). Does it assume how the dynamics model is learned? As the alg can be arbitrary, can I choose an algorithm that does not learn the dynamics so that \sigma_t does not decrease?
- Although Section 4.2 is one of the major contributions of the paper, it is a bit short and dense. For example, Line 254 - 256, a three-line sentence, is a bit hard to parse. Maybe to make room for it, some discussions on the different examples can be moved to the appendix.
- In Page 3 Line 103, it seems that “query the expert” is a stronger assumption than what actually is required: access to expert state trajectories


**Time Spent Reviewing:**

3.5 hours

---

> ### Author Response · Authors · 2021-08-10
> **Response to Reviewer Feedback**
>
> Thank you for your time and feedback.
>
> ## Realizability for cost/transition functions
>
> Note that the realizability of the cost and transition functions can be relaxed to a uniform approximation error, e.g., $\forall s,a$,  $|c(s)-\hat{c}(s)|_\infty < \epsilon_1$, $|P(s,a)-\hat{P}(s,a)|_1 < \epsilon_2$, where, $c$, $P$ are true cost/transition functions and $\hat{c}$, $\hat{P}$ are some cost/transition functions in the function class that we optimize over. The resulting sub-optimality bounds will degrade gracefully, i.e., it will carry factors of $\epsilon_1 \cdot H$ and $\epsilon_2 \cdot H^2$ respectively.
>
> ## Best of T policies vs. last policy
>
> In theory, the best of the T policies, or even, the uniform mixture of the T policies can obtain the guarantees offered by Theorem 3, and yes, in practice, the final policy outputted by the algorithm tends to work well - please refer to figure 1 in the paper (i.e., our learning curve is stable). This distinction between theory and practice occurs in several settings -- even in standard non-smooth convex optimization, standard results for gradient descent require iterate averaging whereas, in practice, the final iterate tends to work well.
>
> ## Dependence on H compared to FAIL
>
> In terms of the statistical error (i.e. dependence on number of samples from the expert policy given to the algorithm), both FAIL and MobILE have the same H dependence. That said, FAIL can only guarantee to work with discrete actions whereas MobILE can work for problems with both discrete/continuous actions.
>
> ## N expert states per iteration
>
> This is written in algorithm 1 for simplicity of analysis. Modifying the theorem/analysis without that step is relatively straightforward as well (i.e., require a union bound all over iterations).  In practice, as we did in experiments, all we require is a fixed dataset of states visited by the expert policy for running the algorithm to produce the results in the paper.
>
> ## Information Gain and model learning
>
> Note that our setting is that we learn a model and its uncertainty measure $\sigma$ over the trajectories generated by the Alg. Since we train the model over the generated trajectories, and the information gain is measuring the model uncertainty over the same training data, we can expect this quantity to be under control (i.e., they grow sublinearly with respect to T), conditioned on the constructed uncertainty measure $\sigma$ being correct. This is of course an assumption in that if the constructed uncertainty measure is wrong or too loose (e.g., if we set $\sigma$ to be infinite, which is a valid uncertainty measure, but is useless from an algorithmic perspective), such an assumption will not hold. Assumptions like this have been used in many prior works in model-based RL, e.g., [Curi et al. 2020]. We also demonstrate that one can construct a valid uncertainty measure $\sigma$ such that this assumption holds in the examples we included in the paper (furthermore, the seminal work [Srinivas et al. 2010] shows that this assumption holds for many commonly used kernels), and in experiments we further empirically verified that this quantity decays when $\sigma$ is approximated by the model ensemble disagreement.
>
> ## Section 4.2, Writing suggestions, Definition of Regret
>
> Thanks for these suggestions, we will incorporate this feedback. We will define regret earlier in the problem setup.
>
> ## References
>
> [Curi et al. 2020] Efficient model-based reinforcement learning through optimistic policy search and planning
>
> [Srinivas et al. 2010] Gaussian Process Optimization in the Bandit Setting: No Regret and Experimental Design

---

> > ### Author Response · Authors · 2021-08-18
> > **Updates on Benchmarking : GAIFO and ILPO**
> >
> > # Additional Benchmarking Results
> >
> > Our paper currently compares MobILE against BC, GAIL and BC-O. We note that GAIL outperforms GAILFO across tasks (as noted by several reviewers), and the paper shows MobILE outperforms GAIL and BC-O across tasks considered in the paper. We add two other benchmarks (ILPO and GAIFO) with a view to address reviewer comments:
> >
> > # Comparison against ILPO
> >
> > ILPO [Edwards et al.] is another method for imitation learning from observations that works with MDPs with discrete actions. We compare ILPO against MobILE in Cartpole, Reacher and Swimmer (as noted in the paper, we discretize actions for Reacher and Swimmer).
> >
> > | Environment |     ILPO          | MobILE (ours)     |
> > | ------------------ | ----------------- | --------------------- |
> > | Cartpole-v2    | $430 \pm 5$ |     $500 \pm 0$   |
> > | Reacher-v2    | $-22 \pm 1 $ | $-9.6 \pm 2.05$ |
> > | Swimmer-v2   | $22 \pm 10 $ | $47.7 \pm 5.8$ |
> >
> > We used the authors recommended hyperparameters and set the learned latent action dimension to that of the action dimension and ran the algorithm with an equivalent number of online interactions as MobILE. Note that MobILE is comparable to ILPO in Cartpole and better in both Reacher and Swimmer.
> >
> >
> > # Comparison against GAIFO
> >
> > Below, we present results comparing MobILE against GAIFO [Torabi et al.] on Hopper and Walker (both environments with continuous actions) - we will present results environments on other tasks, as promised in the final version.
> >
> >
> > | Environment |    GAIFO      | MobILE (ours)   |
> > | :----       |    :---:      |        -----:   |
> > | Hopper-v2   |$1700\pm116.32$|$3360.7\pm192.9$ |
> > | Walker2d-v2 |$1350\pm103.02$| $1961.9\pm203.9$|
> >
> > On both Hopper-v2 and Walker2d-v2, we note that MobILE outperforms GAIFO.
> >
> >
> > We hope to engage with the reviewer in a productive manner to bridge any concerns the reviewer has. We would appreciate any communication from the reviewer with a view to improve our paper.
> >
> > ## References
> > [Edwards et al.] Imitating Latent Policies from Observation, ICML 2019.
> >
> > [Torabi et al.] Generative Adversarial Imitation from Observation, ICML Workshop on Imitation, Intent, and Interaction 2019.

---

> > ### Comment · Reviewer_eB6p · 2021-08-21
> > **Thanks to the authors for their reply**
> >
> > Thanks to the authors for their reply. It is very helpful and answers most of my questions. Although the writing of the paper can be improved, such as making the answers to my questions clear, and the uniform approximation assumption and $\epsilon H^2$ degradation are not very graceful when the transition function is not realizable, the approach seems novel and the experimental results are promising. I am now leaning toward accepting the paper. I have one more clarification question left.
> >
> > ## N expert states per iteration ##
> >
> > This question is important as it implies whether the proposed algorithm needs TN or N expert samples, where T is the number of iterations and N is the number of expert samples used in each iteration. If I understand the authors’ reply correctly, although the algorithm needs TN expert samples, the analysis can be easily modified so that N expert samples are needed. If so, I would suggest modifying the algorithm and proof.

---

> > > ### Author Response · Authors · 2021-08-22
> > > **Thank you for your constructive feedback and response!**
> > >
> > > Thank you for your positive feedback about the paper and your constructive comments.
> > >
> > > ## Regarding N expert samples
> > >
> > > Yes indeed, the algorithm just needs to use N fixed expert samples, i.e., there is no need to draw fresh samples per iteration. We realize that our analysis has already taken care of that, i.e., in lemma 9 in appendix, we have already taken a union bound over all iteration t. Our current analysis indeed just works for a fixed expert dataset with N expert samples. Our experimental results also uses a fixed expert dataset. We just need to revise our pseudo-code in the final version to reflect appropriately.

---

### Official Review · Reviewer_fGHV · 2021-07-16

**Rating:** 4
**Confidence:** 3

**Summary:**

This paper studies in the setting of Imitation Learning from Observations (ILFO).
The contribution is proposing a framework for policy learning with uncertainty estimates by dynamics model learning, which leads to better balanced exploration and imitation.


**Limitations And Societal Impact:**

This paper tackles sample inefficiency in ILFO settings, which is a key problem in the previous studies. However, as mentioned in the main review, the paper does not discuss well on its applicability, which potentially could be the limitation of the proposed method.


**Main Review:**


The paper proposes a framework for imitation learning from observations, MobLE incorporating policy learning with exploration bonus by learned dynamics model. The framework of MobILE seems reasonable for the motivation and general one, however, there seem to be some concerns on the discussion of the motivation of this study, and explanation of the method and experiments.

## Pros
* Proposing a unified framework for ILFO with a combination of policy learning, exploration bonus measured with a learned model ensemble. The combination is novel in my understanding and potentially applicable to other algorithm choices.
* Giving a mathematical explanation for why the proposed exploration strategy is more efficient than random (or no) exploration.

## Concerns
### Discussion on the applicability of the work
The key idea of MobILE is dependent on exploration bonus with disagreement between learned dynamics model ensemble. I have questions about how this bonus is generalizable to other settings (e.g. input) than the ones shown in the experiments.

The experiments in the paper are performed on “state” input, and the learned policies are tested on the same robot (MDP) as the expert. For me, the assumption of no access to the actions of an expert is tricky and overly simplified in this case, since we have access to actions of the learned policy at the same time. Giving the possible situations where the assumption holds will make the paper more convincing.

Moreover, in the experiments in Section 6, the bonus is measured with L2 norm of state space, however, in the practical applications, the estimate seems not straightforward. For example, learning from video experts is a promising application, since videos of performing a task are easier to obtain. But how can we measure the bonuses if we use videos as the experts without actions? There is some literature on imitation learning from video experts [1-4], and is it possible to incorporate MobiLE framework with these visual methods?

### Comparison with other ILFO methods
In the experiments, the proposed method is compared against BC, BC-O, and GAIL. In these methods, only BC-O is ILFO method except for the proposed one. However, there are some previous methods in ILFO with different characteristics as mentioned in Section 1.1. What is the empirical difference between the proposed framework and the previous one (e.g. estimating inverse dynamics)? Adding comparison with other ILFO methods will convince the reader how exploration in the MobLE framework is promising.

### Number of the models in the experiment and the formulation of uncertainty (L165)
In L165, the uncertainty is defined with "sup" of L2 norm between two different models from possible pairs in the ensemble (termed as “maximum disagreement” in L219). However, the experiments shown in the paper are performed only with two dynamics models (the uncertainty is purely measured with discrepancy of next state predictions between two models). So, there is a mismatch between the formulation and experiment. L276-277 mentions that using more than two models does not improve performance much, and I suppose these results should be included in Appendix D for justification of the formulation of taking “sup” of mismatches between pairs of models.

## Additional Comments
### Model learning for optimizing the policy
In the proposed algorithm, the dynamics model learning is used only for calculating the exploration bonus, which I feel is a bit costly for its computation, especially in settings with high-dimensional states. Is it possible to use them to boost policy learning as well, like [5-6]?

### Minor correction
* L298: BC-0 -> BC-O


[1] K. Schmeckpeper, et al. Reinforcement Learning with Videos: Combining Offline Observations with Interaction. https://arxiv.org/abs/2011.06507

[2] A. Edwards, et al. Imitating Latent Policies from Observation. https://arxiv.org/abs/1805.07914

[3] K. Schmeckpeper, et al. Learning Predictive Models From Observation and Interaction. https://arxiv.org/abs/1912.12773

[4] M. Chang, et al. Semantic Visual Navigation by Watching YouTube Videos. https://arxiv.org/abs/2006.10034

[5] S. Sutton, et al. (1991). ​​Dyna, an integrated architecture for learning, planning, and reacting.

[6] T. Kurutach, et al. Model-Ensemble Trust-Region Policy Optimization.
https://arxiv.org/abs/1802.10592


**Time Spent Reviewing:**

6

---

> ### Author Response · Authors · 2021-08-10
> **Response to Reviewer Feedback**
>
> Thank you for your time and feedback.
>
> ## Applicability of the work
>
> With regards to vision based tasks (learning from video experts): There have been advances in learning dynamics models for these classes of problems, e.g. the Dreamer-v2 model [Hafner et al 2020]; in these contexts, disagreement between models for bonuses can be measured in a latent space. For instance, see [Rafailov et al. 2021] for using disagreement in latent space for setting penalty in offline RL; one can utilize a similar strategy to instead define bonuses for exploration within MobILE. However we would like to point out that not every control problem needs to be framed as pixel-to-torque control. Indeed, model-based RL under compact states (not images) is still being widely studied in the RL community as well as in imitation learning (as a representative example, see GAIL [Ho & Ermon 2016]), and in ILfO (see BC-O [Torabi et al. 2018], FAIL [Sun et al. 2019], GAIFO [Torabi et al. 2018] and many others).
>
> Aside from extension to pixel-based states, we would like to point out that the setting we considered in this paper has been well defined and explored in many published prior works. Second, there are indeed interesting applications where the ILFO framework can be directly applied. For instance, Peng et al SIGGRAPH 21 and Peng et al RSS 20 developed pipelines which convert motion capture data into state-wise demonstrations in simulation where states are compact states of the simulated robots rather than images. After such conversion, it is exactly the ILFO setting we considered here!
>
> We’d be more than happy to add discussion surrounding applicability of MobILE to these problem settings, but, emphasize that ILfO with benchmarks utilized by this paper has a great deal of precedence in prior works, both in ILfO and IL.
>
> ## Comparison with other ILFO works
>
> We ran GAIFO on Walker2d-v2 during the rebuttal period and we note that GAIFO achieves a best performance of 1350 (averaged across seeds), which corresponds to a normalized score of 0.675 and this is lower than both GAIL (which gets near 1600, normalized score = 0.8) and MobILE (which achieves expert performance of 2000, normalized score = 1) - see the bar plot in Figure 1.
>
> We’d be happy to include GAIFO’s results in the paper, but, we wish to emphasize that GAIFO is outperformed by GAIL, which in turn is outperformed by MobILE. Furthermore, as mentioned in the paper, we use GAIL as a surrogate for GAIFO’s performance, which is clear since GAIFO’s result (even from their own paper) is sub-optimal compared to GAIL. With regards to comparison to ILPO and FAIL, we will commit to presenting these benchmarks on the subset of discrete action tasks considered in the paper - but, we wish to make a note that MobILE is applicable to both discrete and continuous action spaces while FAIL/ILPO can work with only discrete action spaces.
>
>
> ## Number of models in ensemble, formulation of uncertainty
>
> With regards to two models in ensemble - note that prior work such as Osband et al. 2018 does show that an ensemble with two models works well for setting exploration bonuses, and there are modest gains to be had with larger number of models in the ensemble. In terms of relative benefit of how much computation effort we spend to maintain/update the ensemble compared to the benefits that a larger ensemble offers in terms of estimated bonuses, we found even an ensemble with two models suffices to obtain the performance as shown in the experiments. We will present an ablation study on the effect of the number of models in the ensemble on the performance of MobILE. But we emphasize that our ablation study shows that bonus does make a difference even with two models in the ensemble.
>
>
> ## Model Learning for optimizing the policy
>
> Yes, as the title of our paper indicates - we perform policy learning within the learnt model - in particular, we utilize rollouts within the models in the ensemble. Model-based optimization is indeed one of the key reasons behind MobILE’s sample efficiency.
>
> ## References
>
> [Hafner et al. 2020] Mastering Atari with Discrete World Models
>
> [Rafailov et al. 2021] Offline Reinforcement Learning from Images with Latent Space Models
>
> [Peng et al. SIGGRAPH 2021] AMP: Adversarial Motion Priors for Stylized Physics-Based Character Control
>
> [Peng et al. RSS 2020] Learning Agile Robotic Locomotion Skills by Imitating Animals

---

> > ### Author Response · Authors · 2021-08-18
> > **Updates on Benchmarking : GAIFO and ILPO**
> >
> > # Additional Benchmarking Results
> >
> > Our paper currently compares MobILE against BC, GAIL and BC-O. We note that GAIL outperforms GAILFO across tasks (as noted by several reviewers), and the paper shows MobILE outperforms GAIL and BC-O across tasks considered in the paper. We add two other benchmarks (ILPO and GAIFO) with a view to address reviewer comments:
> >
> > # Comparison against ILPO
> >
> > ILPO [Edwards et al.] is another method for imitation learning from observations that works with MDPs with discrete actions. We compare ILPO against MobILE in Cartpole, Reacher and Swimmer (as noted in the paper, we discretize actions for Reacher and Swimmer).
> >
> > | Environment |     ILPO          | MobILE (ours)     |
> > | ------------------ | ----------------- | --------------------- |
> > | Cartpole-v2    | $430 \pm 5$ |     $500 \pm 0$   |
> > | Reacher-v2    | $-22 \pm 1 $ | $-9.6 \pm 2.05$ |
> > | Swimmer-v2   | $22 \pm 10 $ | $47.7 \pm 5.8$ |
> >
> > We used the authors recommended hyperparameters and set the learned latent action dimension to that of the action dimension and ran the algorithm with an equivalent number of online interactions as MobILE. Note that MobILE is comparable to ILPO in Cartpole and better in both Reacher and Swimmer.
> >
> >
> > # Comparison against GAIFO
> >
> > Below, we present results comparing MobILE against GAIFO [Torabi et al.] on Hopper and Walker (both environments with continuous actions) - we will present results environments on other tasks, as promised in the final version.
> >
> >
> > | Environment |    GAIFO      | MobILE (ours)   |
> > | :----       |    :---:      |        -----:   |
> > | Hopper-v2   |$1700\pm116.32$|$3360.7\pm192.9$ |
> > | Walker2d-v2 |$1350\pm103.02$| $1961.9\pm203.9$|
> >
> > On both Hopper-v2 and Walker2d-v2, we note that MobILE outperforms GAIFO.
> >
> >
> > We hope to engage with the reviewer in a productive manner to bridge any concerns the reviewer has. We would appreciate any communication from the reviewer with a view to improve our paper.
> >
> > ## References
> > [Edwards et al.] Imitating Latent Policies from Observation, ICML 2019.
> >
> > [Torabi et al.] Generative Adversarial Imitation from Observation, ICML Workshop on Imitation, Intent, and Interaction 2019.

---

### Official Review · Reviewer_ji99 · 2021-07-17

**Rating:** 6
**Confidence:** 2

**Summary:**

This paper studies a model-based approach to imitation learning from observation (ILFO). The main idea is to learn a forward model as well as ‘strategic’ exploration which is based on the concept of optimism in face of uncertainty. The proposed algorithm, MobILE, combines (i) model-based RL (forward dynamics learning), (ii) exploration based on uncertainty (maximum disagreement of model), and (iii) imitation learning. More specifically, models and the novelty-bonus function learned from online interaction with the environment can be plugged in to optimize the IPM imitation learning objective, where the bonus term incorporated into the discriminator would diminish the discriminator’s power for novel states. This algorithm features an ability to to trade off exploration and exploitation(imitation), and achieves good results on standard gym continuous control benchmarks (as good as imitation learning with action data available) despite the challenging ILFO setting.

**Limitations And Societal Impact:**

Section 7 discusses limitation of the work a little bit, but more detailed or comprehensive discussion would be also worth considering. There is concerns or issues regarding societal impact.

**Main Review:**

Overall, I think this work is a well-written paper with solid theoretical contribution and strong empirical result. The main contribution is to present a novel combination of model-based imitation learning and exploration (uncertainty and optimism), and to show theoretical optimality bound as well as empirical validation. Given its novelty and contribution, I am slightly leaning towards recommending acceptance at this point.

Significance: The proposed approach seems to be designed with principle and makes sense. The MobILE framework has some good advantages compared to existing work such as FAIL and ILPO, such as it is compatible with continuous control and does not require much cost function tuning. Actually the practical algorithm is quite a straightforward combination of model/exploration/GAIL component, but some theoretical insights provided would be a novel and significant result within the ILFO setup. Although I was unable to examine and validate all theoretical results and proofs, the theoretical results on optimality bound seems significant and important. It is interesting to see that the difficulty of ILFO over standard IL is formalized through the exponential gap, which is also a good theoretical contribution.

Regarding the exploration bonus $b$, don’t you need to have a weight (either as a hyperparameter or a learnable variable) to balance the impact of exploration? It is obvious that such exploration terms can improve the algorithm’s performance (as shown in Section 6.2), but how important are they and to what extent do they contribute to the imitation learning?

Regarding experiments, one concern I have is that a direct comparison to sensible baselines (GAIfO [64]) is lacking. Although it is reported that GAIL outperfoms GAIfO in all tasks (which is very expected), it would have been much better to have a direct comparison between MobILE and GAIfO or other baseline. For example, is exploration mostly accountable for the performance improvement under the same “learning from observation” setting? Without optimism (i.e. just model-based GAIfO) would it still work better or comparably to those LfO approaches?

A minor question) why does BC-O perform much better than BC in CartPole? Is it just a mistake? In L293, it is said that BC outperforms BC-O in all benchmark tasks.


Writing/Clarity:
- Motivation of the problem is well-written, and discussion of related work looks comprehensive and informative.
- With its theoretical contribution in mind, the paper is overall a bit difficult to follow. As a reviewer and for readers who do not have much expertise on imitation learning and theoretical RL, I wish there were a more kind background or preliminaries section (around section 2. Setting) to review the GAIL and IPM setup for the sake of self-containedness and more clarity.
- Naming of the problem: I think LfO is a more common, popular, established abbreviation to denote learning from observation. In that regard, why don’t authors use the term ILfO instead of ILFO? I also suggest the authors to refer GAIFO as “GAIfO” as per the original paper [64].
- L107: It would be good to write down the optimization objective that includes $c$ (the cost function) for better clarity and how it connects to the IPM objective (L117). Given that $V^\pi$ is the expected total cost under $c$, it’d be good to have a more formal and clear definition.



**Time Spent Reviewing:**

10+

---

> ### Author Response · Authors · 2021-08-10
> **Response to Reviewer Feedback**
>
> Thank you for your time and your feedback.
>
> ## Bonus scaling
>
> Yes, we do scale the magnitude of bonuses - for instance, see lines 275-280 in the main paper and section C.2.5 in the appendix where we indicate that we use a tunable parameter \lambda to scale the bonus relative to the discriminator costs. Tuning the bonus is equivalent to tuning the tradeoff between imitation and exploration.
>
> ## Benchmarking
>
> Thank you for noting that GAIFO is actually outperformed by GAIL. We ran GAIFO on Walker2d-v2 during the rebuttal period and we note that GAIFO achieves a best performance of 1350 (averaged across seeds), which corresponds to a normalized score of 0.675 and this is lower than both GAIL (which gets near 1600, normalized score = 0.8) and MobILE (which achieves expert performance of 2000, normalized score = 1) - see the bar plot in Figure 1.
>
> We’d be happy to include GAIFO’s results in the paper, but, we wish to emphasize that GAIFO is outperformed by GAIL, which in turn is outperformed by MobILE. Furthermore, as mentioned in the paper, we use GAIL as a surrogate for GAIFO’s performance, which is clear since GAIFO’s result (even from their own paper) is sub-optimal compared to GAIL. With regards to comparison to ILPO and FAIL, we will commit to presenting these benchmarks on the subset of discrete action tasks considered in the paper - but, we wish to make a note that MobILE is applicable to both discrete and continuous action spaces while FAIL/ILPO can work with only discrete action spaces.
>
> ## Ablation studies
>
> Ablation without optimism - note that we perform ablation studies for all our environments for MobILE with and without optimism (cf. figure 2).
>
> What happens to GAIFO with optimism? Note that this is an interesting question to explore as a followup to the GAIFO, and we’d like to point out that this isn't part of this paper's contributions (i.e., how to incorporate bonus into a model-free IL algorithm in a principled and provable manner is an interesting research question by itself). That said, our conjecture is that MobILE will be sample efficient compared to GAIFO with optimism because MobILE is model-based whereas GAIFO with optimism (similar to GAIL) is a model-free algorithm.
>
> ## BC-O's behavior
>
> We also noticed this result when running the experiment - and we have run tests with this code (and of course tried our best tuning parameters for BC as well) and observed that averaged across many seeds, BC-O does outperform BC for cartpole.
>
> ## Writing/Clarity
>
> Thank you for the suggestions, and we will address these comments.

---

> > ### Author Response · Authors · 2021-08-18
> > **Updates on Benchmarking : GAIFO and ILPO**
> >
> > # Additional Benchmarking Results
> >
> > Our paper currently compares MobILE against BC, GAIL and BC-O. We note that GAIL outperforms GAILFO across tasks (as noted by several reviewers), and the paper shows MobILE outperforms GAIL and BC-O across tasks considered in the paper. We add two other benchmarks (ILPO and GAIFO) with a view to address reviewer comments:
> >
> > # Comparison against ILPO
> >
> > ILPO [Edwards et al.] is another method for imitation learning from observations that works with MDPs with discrete actions. We compare ILPO against MobILE in Cartpole, Reacher and Swimmer (as noted in the paper, we discretize actions for Reacher and Swimmer).
> >
> > | Environment |     ILPO          | MobILE (ours)     |
> > | ------------------ | ----------------- | --------------------- |
> > | Cartpole-v2    | $430 \pm 5$ |     $500 \pm 0$   |
> > | Reacher-v2    | $-22 \pm 1 $ | $-9.6 \pm 2.05$ |
> > | Swimmer-v2   | $22 \pm 10 $ | $47.7 \pm 5.8$ |
> >
> > We used the authors recommended hyperparameters and set the learned latent action dimension to that of the action dimension and ran the algorithm with an equivalent number of online interactions as MobILE. Note that MobILE is comparable to ILPO in Cartpole and better in both Reacher and Swimmer.
> >
> >
> > # Comparison against GAIFO
> >
> > Below, we present results comparing MobILE against GAIFO [Torabi et al.] on Hopper and Walker (both environments with continuous actions) - we will present results environments on other tasks, as promised in the final version.
> >
> >
> > | Environment |    GAIFO      | MobILE (ours)   |
> > | :----       |    :---:      |        -----:   |
> > | Hopper-v2   |$1700\pm116.32$|$3360.7\pm192.9$ |
> > | Walker2d-v2 |$1350\pm103.02$| $1961.9\pm203.9$|
> >
> > On both Hopper-v2 and Walker2d-v2, we note that MobILE outperforms GAIFO.
> >
> >
> > We hope to engage with the reviewer in a productive manner to bridge any concerns the reviewer has. We would appreciate any communication from the reviewer with a view to improve our paper.
> >
> > ## References
> > [Edwards et al.] Imitating Latent Policies from Observation, ICML 2019.
> >
> > [Torabi et al.] Generative Adversarial Imitation from Observation, ICML Workshop on Imitation, Intent, and Interaction 2019.

---

### Official Review · Reviewer_iytC · 2021-07-21

**Rating:** 6
**Confidence:** 3

**Summary:**

This paper introduces MobILE, an approach for training agents to imitate from expert states only, i.e., when actions are unavailable. In particular, the approach utilizes an exploration bonus to more efficiently explore within the environment. The approach is compared to standard imitation learning approaches within OpenAI gym tasks.


**Limitations And Societal Impact:**

The authors discuss some potential societal impacts, particularly that it is import to ensure the quality of expert policies.

**Main Review:**

Clarity: Overall, I found the paper to be clearly motivated and mostly well-written, though I found section 2.1 to be somewhat confusing.

Originality: This approach introduces using exploration with an imitation from observation approach. Adversarial methods have been used for training both Imitation Learning (IL) and Imitation from Observation (IfO). The main contribution is using an additional exploration bonus to train an IfO method to tradeoff exploration and imitation.

Quality: I would have liked to see more ablations on each individual contribution of the paper. For example, the original GAIL-style training compared to the IPM method used in the paper (without bonus) and GAIL with the exploration bonus compared to MobILE. If this would not be straightforward then other exploration methods could perhaps also be considered.

In general, I am unsure if this work is a contribution to the IfO or IL community, particularly because the forward-dynamics model is learned online and hence does not seem specific to training IfO methods. It would be great if the authors could comment on how this approach is a contribution to IfO.

Significance: The reason I am leaning towards rejecting the paper is that I am unsure of the significance of the approach. It seems the main contribution is to utilize a better exploration approach to imitation from observation than random exploration. This could be useful, as exploration has not been well-studied in the imitation from observation literature. However, the work is not compared to any other exploration methods. Since the exploration bonus is learned from data captured online, other intrinsic exploration bonuses used for RL, such as curiosity [3] or RND [4] could have been evaluated. The bonus used in the paper is in fact quite similar to that used in [3] and so it would be good to at least include it in the related work section. Additionally, it is unclear how much of a contribution using ensembles for the bonus is given the paper mentions having more than two ensembles did not lead to any further gains.

Another problem is that the experiments are somewhat weak. The approach is only compared to a single IfO approach, BCO, but there have been several advances since this work was introduced. While the paper discusses other such approaches such as FAIL [5] and ILPO [6], the argument is that these approaches only work in environments with discrete actions, but MobILE also works with discrete actions and so it should be reasonable to compare in these settings.

Comments:

- The adversarial setting (i.e. the purpose of discriminators) should be more clearly and formally explained.
- Could the authors further discuss why a forward dynamics model is better defined than an inverse dynamics model? If the dynamics are stochastic for example then $\Vert f(s,a) - s’ \Vert$ $f$ will just learn to predict the expected $s’$ rather than a deterministic example. But this is also the case in $\Vert I(s,s’) - a \Vert$ if there is not a unique $a$. I.e. it will just compute the expected $a$ rather than a deterministic one. Practically, I do not imagine many environments where this would be much of an issue.
- Why is BCO constant in the figures? It also has an online stage where the inverse dynamics model is being learned and getting increasingly better experiences to train the bc policy. How were the dynamics models trained?
- I was surprised to see that the approach performs better than GAIL, since that method has ground-truth actions. Could the authors discuss this result in more detail?
- Was it necessary to scale/bound the outputs of the exploration bonus?
- The graphs should include tick marks on the plots (rather than only color) for better accessibility

[1] Generative Adversarial Imitation Learning. Ho et al.

[2] Generative Adversarial Imitation Learning from Observation. Torabi et al.

[3] Curiosity-driven Exploration by Self-supervised Prediction. Pathak et al.

[4] Exploration by Random Network Distillation. Burda et al.

[5] Provably Efficient Imitation Learning from Observation Alone. Sun et al.

[6] Imitating Latent Policies from Observation. Edwards et al.

**Time Spent Reviewing:**

5

---

> ### Author Response · Authors · 2021-08-10
> **Response to Reviewer Feedback**
>
> Thank you for your time and feedback.
>
> ## Other exploration methods
>
> This paper establishes why exploration is an important component of ILfO and presents an algorithmic framework that incorporates exploration within a distribution matching framework. To the best of our knowledge, these contributions are new to the ILfO literature. Note that corresponding to our main contribution, we have conducted ablations with and without optimism and our experiments (in figure 2) suggest optimism is indeed crucial to improving performance distribution matching algorithms for ILfO.
>
> We agree that a systematic treatment of other exploration methods is an interesting topic for follow up works, particularly when expanding the scope of problems that ILFO can be applied to (e.g. with high-dimensional problems like videos/images). Note that settings with compact state description has featured heavily in the literature of RL, Imitation Learning (as a representative example, see GAIL [Ho & Ermon 2016]), and Imitation Learning from Observation Alone (see BC-O [Torabi et al. 2018], FAIL [Sun et al. 2019], GAIFO [Torabi et al. 2018] and many others).
>
> We have already cited the paper of Pathak et al., but, we will present this in the related work.
>
> ## Why Ensembles for bonus
>
> Ensembles have been widely utilized for bonus based methods in RL, e.g., Osband et al. 2018, Azizzadenesheli et al. 2018, Pathak et al. 2019, Lowrey et al. 2019 (all cited in the paper) - these methods have shown great degree of success in practice. Moreover, ensembles connect with the theoretical results presented in the paper, see example 3 in page 6.
>
> ## Why only two models in ensemble
>
> Prior work such as Osband et al. 2018 does show that an ensemble with two models often works well already, and there are modest gains to be had with larger numbers of models in the ensemble. We will present an ablation study to characterize the effect of the number of models in the ensemble on the performance of MobILE. But we emphasize that our ablation studies during running experiments for the paper indicated that disagreement between 2 models in the ensemble already gives good results compared to using no bonus within the distribution matching procedure.
>
> ## Benchmarks
>
> We ran GAIFO on Walker2d-v2 during the rebuttal period and we note that GAIFO achieves a best performance of 1350 (averaged across seeds), which corresponds to a normalized score of 0.675 and this is lower than both GAIL (which gets near 1600, normalized score = 0.8) and MobILE (which achieves expert performance of 2000, normalized score = 1) - see the bar plot in Figure 1.
>
> We’d be happy to include GAIFO’s results in the paper, but, we wish to emphasize that GAIFO is outperformed by GAIL, which in turn is outperformed by MobILE. Furthermore, as mentioned in the paper, we use GAIL as a surrogate for GAIFO’s performance, which is clear since GAIFO’s result (even from their own paper) is sub-optimal compared to GAIL. With regards to comparison to ILPO and FAIL, we will commit to presenting these benchmarks on the subset of discrete action tasks considered in the paper - but, we wish to make a note that MobILE is applicable to both discrete and continuous action spaces while FAIL/ILPO can work with only discrete action spaces.
>
> ## Discriminators
>
> These are functions that help define costs used by a planning algorithm (TRPO/NPG) for purposes of distribution matching with the expert states. We will explain this in detail.
>
> ## Forward vs. Inverse Dynamics model
>
> Like mentioned in introduction and related work in the paper, note that forward dynamics are well defined for MDPs whereas, inverse dynamics are applicable only for classes of MDPs with injective dynamics [Zhu et al. NeurIPS 2020]. Please refer to remark 1, section 9.3 of Zhu et al. (NeurIPS 2020) for details on the limitations of algorithms relying on inverse dynamics compared to ones relying on forward dynamics models.
>
> One way to note why inverse dynamics models are ill-defined is through the Bayes rule - we have P(a | s,s’) $\propto$ p(s) \pi(a|s) P(s’ | s,a), i.e., an inverse dynamics is associated with a policy and a prior over s. Thus, during learning, when policy is being updated, the inverse dynamics is changing as well (e.g., imagine a replay buffer containing samples collected from many different policies, it’s unclear what the trained inverse model is). In contrast, forward dynamics model is well defined and does not change during learning.  Said though, we agree that in practice training inverse model tends to work, especially when the dynamics is injective and deterministic [Zhu et al. 2020].
>
> ## BC-O's performance
>
> We plot the policy which obtains the best performance by running BC-O for the given online sample budget; this tends to benefit BC-O’s result. In many of the environments considered in this paper, by giving this advantage to BC-O, we still note that it performs highly sub-optimally compared to other algorithms that we benchmark MobILE against.
>
> ## MobILE vs GAIL
>
> Note that GAIL is a model free method whereas MobILE is model-based and thus inherits the sample complexity benefits offered by model-based methods over model-free methods.
>
> ## Bonus scaling
>
> Yes, we do scale the magnitude of bonuses - for instance, see lines 275-280 in the main paper and section C.2.5 in the appendix where we indicate that we use a tunable parameter \lambda to scale the bonus relative to the discriminator costs. Tuning the bonus is equivalent to trading-off between imitation and exploration.
>
> ## Comments about the plots
>
> We will make these changes, thank you for your suggestions.
>
> ## References
> [Zhu et al. 2020] Off-policy imitation learning from observation.

---

> > ### Author Response · Authors · 2021-08-18
> > **Updates on Benchmarking : GAIFO and ILPO**
> >
> > # Additional Benchmarking Results
> >
> > Our paper currently compares MobILE against BC, GAIL and BC-O. We note that GAIL outperforms GAILFO across tasks (as noted by several reviewers), and the paper shows MobILE outperforms GAIL and BC-O across tasks considered in the paper. We add two other benchmarks (ILPO and GAIFO) with a view to address reviewer comments:
> >
> > # Comparison against ILPO
> >
> > ILPO [Edwards et al.] is another method for imitation learning from observations that works with MDPs with discrete actions. We compare ILPO against MobILE in Cartpole, Reacher and Swimmer (as noted in the paper, we discretize actions for Reacher and Swimmer).
> >
> > | Environment |     ILPO          | MobILE (ours)     |
> > | ------------------ | ----------------- | --------------------- |
> > | Cartpole-v2    | $430 \pm 5$ |     $500 \pm 0$   |
> > | Reacher-v2    | $-22 \pm 1 $ | $-9.6 \pm 2.05$ |
> > | Swimmer-v2   | $22 \pm 10 $ | $47.7 \pm 5.8$ |
> >
> > We used the authors recommended hyperparameters and set the learned latent action dimension to that of the action dimension and ran the algorithm with an equivalent number of online interactions as MobILE. Note that MobILE is comparable to ILPO in Cartpole and better in both Reacher and Swimmer.
> >
> >
> > # Comparison against GAIFO
> >
> > Below, we present results comparing MobILE against GAIFO [Torabi et al.] on Hopper and Walker (both environments with continuous actions) - we will present results environments on other tasks, as promised in the final version.
> >
> >
> > | Environment |    GAIFO      | MobILE (ours)   |
> > | :----       |    :---:      |        -----:   |
> > | Hopper-v2   |$1700\pm116.32$|$3360.7\pm192.9$ |
> > | Walker2d-v2 |$1350\pm103.02$| $1961.9\pm203.9$|
> >
> > On both Hopper-v2 and Walker2d-v2, we note that MobILE outperforms GAIFO.
> >
> >
> > We hope to engage with the reviewer in a productive manner to bridge any concerns the reviewer has. We would appreciate any communication from the reviewer with a view to improve our paper.
> >
> > ## References
> > [Edwards et al.] Imitating Latent Policies from Observation, ICML 2019.
> >
> > [Torabi et al.] Generative Adversarial Imitation from Observation, ICML Workshop on Imitation, Intent, and Interaction 2019.

---

> > > ### Comment · Reviewer_iytC · 2021-08-23
> > > **Response to comments**
> > >
> > > Thank you for the comments and additional experiments. The additional comparisons should improve the quality of the paper. I would still appreciate some discussion on how this work is a contribution to IfO. In particular, could this same method be applied to imitation learning and if so was there a particular reason to focus on IfO?
> > >
> > > Additionally, what was the exploration method used in the GAIL/GAIfO baselines? Was it just adding noise to the policy? And how was the data collected for training the inverse dynamics model in BCO? In particular, was the data collected using a random policy or the policy that was being learned?

---

> > > > ### Author Response · Authors · 2021-08-24
> > > > **Thank you for your response.**
> > > >
> > > > Thank you for your message and feedback on the paper.
> > > >
> > > > # Can MobILE be utilized for Imitation Learning (IL)?
> > > >
> > > > MoBILE could definitely be applied to IL, but we want to emphasize that with access to expert actions, strategic exploration as done in MoBILE is not necessary. In Section 4.2, we gain two key insights from the lower bound:
> > > >
> > > > ## 1. IfO can be solved more efficiently by using strategic exploration as opposed to using random exploration
> > > >
> > > > In IL, expert actions serve as a supervision signal for learning a policy that imitates the expert and a reduction to a supervised learning approach already works well in theory. It is unclear if strategic exploration is necessary in IL. In IfO, since we do not observe actions, we need to try different actions to see if the outcome matches the observed expert states. We show how this trial and error procedure can be sped up by utilizing strategic exploration as opposed to using random exploration or no exploration. To the best of our knowledge, existing works in IfO haven’t considered the explicit use or benefits of strategic exploration for the IfO problem. Our paper shows the benefits of strategic exploration (both in theory and practice) for solving the IfO problem.
> > > >
> > > > ## 2. There is an exponential gap in terms of sample complexity between IfO and IL
> > > >
> > > > In sec 4.2, we show there exists an MDP, where any IfO algorithm will suffer $\Omega(\sqrt{\frac{A}{T}})$ average regret (with strategic exploration, mobile matches to this lower bound, while random exploration based strategy will have sub-optimal regret of $O(T^{-1/3})$). However, if the algorithm were presented with expert actions, there is an IL algorithm that can achieve $O(\sqrt{\frac{\ln(A)} {T}})$ rate on the same MDP. This exponential gap in terms of dependence on A between IfO and IL highlights that IfO is fundamentally harder than IL.
> > > >
> > > > We chose to focus on IfO over IL to investigate how to solve IfO more efficiently by strategic exploration. We show in Section 6.2, that our designed exploration method solves IfO more efficiently than random exploration. It is an interesting future direction to investigate if strategic exploration is necessary and useful in IL from both practical and theoretical perspectives.
> > > >
> > > > # Exploration Method used in GAIL/GAIFO/BCO
> > > >
> > > > Both GAIL and GAIFO use TRPO as their default policy gradient algorithm. For both baselines, we use the default TRPO policy parameterizations where exploration is handled by sampling an action from a gaussian policy head.
> > > >
> > > >
> > > > # Data collection for inverse dynamics learning in BCO
> > > >
> > > > The data was collected using the policy that was being learned. Specifically, for continuous action spaces, we learn a policy with a gaussian head; while for discrete action spaces, the policy computes the probability of an action by a softmax over the actions.  BCO interchanges inverse dynamics learning and policy learning where samples from the updated policy are then used to improve the inverse dynamics model. For the exact hyperparameters, we chose the recommended hyperparameters presented by BCO’s paper/codebase.
> > > >
> > > >
> > > > Let us know if there are any other questions/points of discussion.

---

> > > > > ### Comment · Reviewer_iytC · 2021-08-24
> > > > > **Thanks for the additional insights**
> > > > >
> > > > > I now have less doubts about the significance of the approach. I've raised my score.

---

### Decision · Program_Chairs · 2021-09-27

**Decision:**

Accept (Poster)

**Comment:**

I appreciate the authors for detailed rebuttal and additional experimental results, and the reviewers for engaging in detailed feedback (especially reviewers axQZ and eB6p for active discussions). The novelty and impact were better communicated after the discussions. I like that the paper studies a generic problem of ILFO from model-based perspectives, discusses its fundamental differences from IL theoretically, and proposes better strategic exploration through models as a practical solution. One concern is some missing references/baselines for model-based GAIL/GAIFO-like algorithms, e.g. MAIL [Baram et al. 2016], but MobLIE ablation study without optimism appears to cover similar cases.